



# Evaluation of the coupling of EMACv2.55 and the land surface and vegetation model JSBACHv4

Anna Martin[1], Veronika Gayler[2], Benedikt Steil[1], Klaus Klingmüller[1], Patrick Jöckel[3], Holger Tost[4], Jos Lelieveld[1,5], and Andrea Pozzer[1,5]

[1]Max Planck Institute for Chemistry, Atmospheric Chemistry Department, 55128 Mainz, Germany
[2]Max Planck Institute for Meteorology, Climate Dynamics Department, 20146 Hamburg, Germany
[3]Deutsches Zentrum für Luft- und Raumfahrt (DLR), Institut für Physik der Atmosphäre, Oberpfaffenhofen, Germany
[4]Johannes Gutenberg–University Mainz, Institute for Physics of the Atmosphere, 55128 Mainz, Germany
[5]The Cyprus Institute, Climate and Atmosphere Research Center, Nicosia, 1645, Cyprus

**Correspondence:** (a.martin@mpic.de)

**Abstract.** We present the coupling of the Jena Scheme for Biosphere-Atmosphere Coupling in Hamburg (JSBACH) version four to the ECHAM/MESSy Atmospheric Chemistry (EMAC) model. With JSBACH, the soil water bucket model in EMAC is replaced by a diffusive hydrological transport model for soil water that includes water storage and infiltration in five soil layers, preventing too rapid soil drying and reducing biases in soil temperature and moisture. A five-layer snow scheme is implemented and phase changes of water in the soil are considered. Multiple land cover types are included to provide a state dependent surface albedo, which accounts for the absorption of solar radiation by vegetation. Plant net primary productivity, leaf area index and surface roughness are calculated according to the plant functional types. This paper provides a detailed evaluation of the coupled model based on observations and reanalysis data, including ERA5/ERA5-Land datasets, Global Precipitation Climatology Project (GPCP) data, and MODIS satellite data. In particular, land surface temperature (LST), terrestrial water storage (TWS), surface albedo ($\alpha$), top of atmosphere radiation flux (Rad$_{\text{TOA}}$), precipitation (precip), leaf area index (LAI), fraction of absorbed photosynthetic active radiation (FaPAR) and gross primary productivity (GPP) are evaluated. The strongest correlation (r) between reanalysis data and the newly coupled model are found for LST ($r = 0.985$, with an average global bias of $-1.546\,K$), $\alpha$ ($r = 0.946$, with an average global bias of $-0.015$) and Rad$_{\text{TOA}}$ ($r = 0.907$, with an average global bias of $3.56, Wm^{-2}$). Precipitation exhibits a correlation with the GPCP dataset of $0.523$ and an average global bias of $0.042\,mm\,day^{-1}$. The LAI climatology in EMAC has been substituted with a refined method for directly calculating LAI. This optimisation yields a correlation of $0.637$ with observations and a global mean deviation of $-0.212$. FaPAR and GPP exemplify two of the many additional variables made available through JSBACH in EMAC. FaPAR and observations show a correlation of $0.663$ with an average global difference of $-0.223$, while the correlation for GPP and observations is $0.564$ and the average global difference is $-0.001\,kg\,Carbon\,km^{-1}$. Benefiting from the numerous added features within the simulated land system, the representation of soil moisture is improved, which is critical for vegetation modelling. This improvement can be attributed to a general increase in soil moisture and water storage in deeper soil layers, leading to a reduction in normalised root mean square error (NRMSE) and a closer alignment of simulated TWS with observations, mitigating the previously widespread problem of soil drought. The correlation of TWS and observations is $0.251$ and the average global difference is $0.052\,m$ water



equivalent. We show that the numerous newly added components strongly improve the land hydrology, e.g. soil moisture, while
surface parameters, which were mostly prescribed according to climatologies, remain similar. The coupling of JSBACH brings
EMAC a step closer towards a holistic comprehensive Earth system model and extends its versatility.

## 1 Introduction

Earth system models (ESM) are needed to analyse current and future climate scenarios, and particularly in view of ongoing
climate change (Masson-Delmotte et al., 2021), it is crucial to include the main Earth system components to identify and
quantify potential feedback mechanisms. These numerical models are based on a mathematical formulation of the physical
and chemical processes, accounting for interactions between the atmosphere, oceans and biogeochemical processes on land
(Flato, 2011). Common ESMs contain an atmospheric general circulation model (A-GCM), an ocean general circulation model
(O-GCM) and a land surface model (LSM). A comprehensive list of existing ESMs can be found, for instance, in Annex II:
Models of the Sixth Assessment Report of the Intergovernmental Panel on Climate Change (Gutiérrez et al., 2021).

The ECHAM5/MESSy Atmospheric Chemistry general circulation model (EMAC) is based on an underlying A-GCM, more
specifically the spectral dynamical core, the large-scale advection, and the „nudging"-method are originally from ECHAM (the
5. generation of "European Centre HAMburg general circulation model" (Roeckner et al., 2006). However, all physical param-
eterisations from ECHAM have been replaced by respective further developed MESSy submodels. Those include a simplified
surface model (SURFACE), the O-GCM MPIOM (Pozzer et al., 2011b) and several other submodels which address atmo-
spheric chemistry, cloud and transport processes (Roeckner et al., 2003; Jöckel et al., 2005, 2010). The coupling is achieved
via the Modular Earth Submodel System (MESSy2) framework, gradually refined and expanded in the past two decades to pro-
vide an infrastructure of submodel and process combinations with a wide range of applications. EMAC is a community model
with a growing number of users contributing to developments in various research areas, e.g. studies on particle concentrations
and aerosols (Kohl et al., 2023; Righi et al., 2023), oxidation capacity (Nussbaumer et al., 2023; Friedel et al., 2023), atmo-
spheric dynamics (Eichinger et al., 2023; Charlesworth et al., 2023) and environmental consequences for human health (Pozzer
et al., 2023; Milner et al., 2023). Furthermore, an alternative dynamic vegetation scheme (LPJ-GUESS) has been coupled to
EMAC, allowing for climate-vegetation interactions, e.g., Vella et al. (2023b) In the following, the implementation and evalu-
ation of the LSM Jena Scheme for Biosphere-Atmosphere Coupling in Hamburg (JSBACH), a substitute of EMAC's current
surface model, is documented. JSBACH is implemented into the MESSy framework following the relevant coding standards.
The dynamical land-surface model JSBACH was first developed as the land model for ECHAM at the Max Planck Institute
for Meteorology (MPI-M) (Reick et al., 2021). Originally it emerged from the combination of all ECHAM5 land processes in
a separate land model and was further developed and refined, now providing a large repertoire of biogeochemical processes
of the ecosystem. The latest version (JSBACHv4) is part of the ICON-LAND model and has not yet been coupled to models
simulating atmospheric chemistry (Pham et al., 2021).

The implementation of JSBACH represents significant progress in the development of the ESM EMAC. As climate change
progresses and the occurrence of extreme weather events increase, the influence of surface processes and vegetation becomes



more prominent (Domeisen et al., 2022). Vegetation and soil water balance are driving factors for surface fluxes of heat and
moisture, affecting temperature, precipitation, atmospheric dynamics and chemistry (Miralles et al., 2019; Lauwaet et al., 2009;
Matyssek et al., 2014; Mellouki et al., 2015). JSBACH replaces the soil water bucket model of SURFACE by a more compre-

hensive five-layer hydrological soil model. This substitution aims at improving the representation of surface energy fluxes of
heat and moisture, reducing biases in surface temperature and subsequent plant stress and its impact on biogenic emissions.
JSBACH enables the analysis of biogeochemical processes on much smaller time scales, including not only climatic scales
but also days and hours. This allows the analysis of the impact of vegetation on atmospheric chemistry, plant stomatal uptake,
volatile organic compound (VOC) emissions and associated feedback mechanisms, and enables a more detailed understanding

of land-atmosphere interactions. Furthermore, the impact on the surface energy budget allows for a more consistent represen-
tation of chemical and transport processes in the atmopsheric boundary layer. In Sect. **2** we give a short description of JSBACH
and document the coupling of EMAC and JSBACH via MESSy, including the tuning of the newly coupled model. In Sect. **3**
the evaluation variables and corresponding observation and reanalysis datasets are introduced. The results and discussion of
the evaluation are presented in Sect. **4**.

## 2    Model description

### 2.1    JSBACH

In this work, we implemented the most recent version of JSBACH (JSBACHv4, Schneck et al., 2022). A detailed description of
the parameterizations used in JSBACH can be found in Reick et al. (2013) and Reick et al. (2021), while Schneck et al. (2022)
present features of the version JSBACHv4 in comparison with the previous one (JSBACHv3), along with an assessment of

the results of both versions. Briefly summarized, on the technical side JSBACHv4 has been improved with modernised source
codes and software infrastructure, while on the application side it offers an improved soil scheme with dynamic calculation
of ground heat conductivity and capacity, taking phase change of water and organic fractions within soil layers into account
(Jungclaus et al., 2022; Schneck et al., 2022; Ekici et al., 2014). It provides a complex soil hydrological transport model
including percolation and storage of water in several soil depths, which gives a realistic estimate of soil desiccation and

corresponding soil temperature and moisture. Additionally, the new version introduces a fractional lake mask, a five-layer snow
scheme, and forest age structures (Schneck et al., 2022; de Vrese et al., 2021; Nabel et al., 2020). The implemented version of
JSBACHv4 does not include natural vegetation dynamics, land-use transitions and the nitrogen cycle from JSBACH3. Those
mechanisms have only recently been adopted to JSBACHv4 and will be added to MESSy in the near future. However, on the
climate time scale, the interactions between climate and vegetation are already available in MESSy through the LPJ-GUESS

interactive vegetation module (Vella et al., 2023a).
       The ICON-Land infrastructure allows a clear separation of the physical processes used in JSBACHv4. The processes used
in this study include vegetation coverage, phenology and plant productivity (defined via gross and net primary productivity
and photosynthesis), a turbulence and radiation scheme, surface energy balance, and exchange fluxes of heat and moisture,
soil and vegetation carbon turnover and disturbances due to wildfires or wind-throw. The processes are listed in **Table 1**. In



JSBACH, subgrid scale heterogeneity is taken into account by a tile approach, i.e., grid boxes are divided into tiles associated
to a specific land cover type (Reick et al., 2021). The concept allows to define processes specific to the different land cover
types. For example processes only related to vegetation (as photosynthesis) are calculated only on vegetated tiles. Based on
Reick et al. (2021), water, carbon, nitrogen, and area are conserved with numerical accuracy. Energy conversion is not yet fully
achieved, since temperature of rain water and heat produced by heterotrophic respiration are not accounted for (Reick et al.,
2021).

To couple JSBACH into EMAC, it is implemented as a new submodel within the MESSy framework, following its well doc-
umented coding standards (Jöckel et al., 2010). Each process of the JSBACHv4 source code (listed in **Table 1**) is implemented
as an individual Fortan module in the MESSy Submodel Core Layer (SMCL) and complemented by a newly created submodel
core layer file. Additionally to the individual jsbach processes the file "*messy_jsbach.f90*" is created, which includes definitions
of land-cover specifications (originally taken from *lctlib_nlct21.def*), tile aggregation subroutines, and the subroutine to read
the JSBACH namelist. This namelist (*jsbach.nml*) serves as a user interface, where input and coupling variables are specified.
The full namelist is available in the **Supplement**; the coupled variables are listed from line 152 to 206 of the namelist. Param-
eters can be defined, and logical switches to modify and adjust the simulation can be set. The module subroutines are called
from a newly created JSBACH interface (*messy_jsbach_si.f90*), which is implemented in the MESSy submodel interface layer
(SMIL). Besides the process calls, the interface includes the creation of new "representations" to expand the EMAC model
grid to new dimensions, for soil-, snow-, and canopy layers and vegetation tiles and the JSBACH output variables are saved
as new "channel objects". Both, "representations" and "channel objects" are elements defined in the submodel CHANNEL,
which handles the memory, data output (including check-pointing) and internal data exchange (Jöckel et al., 2010). JSBACH
was chosen as the LSM for EMAC, since it is already successfully implemented and tested in other models like the Icosahedral
Nonhydrostatic Earth System Model (ICON-ESM) and its predecessor (JSBACH3) in the Max Planck Institute Earth System
Model MPI-ESM 1.2 (Mauritsen et al., 2019), which took part in the Coupled Model Intercomparison Project of phase 5 and 6
Giorgetta et al. (2013). Furthermore, in JSBACH specific ecosystem processes like carbon cycling are included. Those mecha-
nisms will, in combination with an atmospheric chemistry model, provide new and interesting insights into the interactions and
feedback mechanisms between vegetation and atmospheric composition. The combination of EMAC and JSBACH makes it
possible to analyse biogeochemical processes at various spatial and temporal resolutions, from small scale experiments of local
sub-daily effects to global scale climate change experiments, in contrast to the coupling of the dynamic vegetation model LPJ-
GUESS into EMAC (Forrest et al., 2020), in which the vegetation-atmosphere coupling is restricted by the diurnal timestep of
the vegetation scheme.



**Table 1.** JSBACH file overview

| JSBACH Process | MESSy filename (SMCL) | short description |
| --- | --- | --- |
| Fuel | *messy_jsbach_fuel.f90* | Availability of carbon fuel to wildfires. |
| Disturbance | *messy_jsbach_dist.f90* | Carbon relocation due to wind-throw and vegetation fires. |
| Phenology | *messy_jsbach_pheno.f90* | Leaf Area Index and foliage projected cover. |
| Hydrology | *messy_jsbach_hydro.f90* | Soil hydrology. |
| Surface Energy Balance | *messy_jsbach_seb.f90* | Surface latent and sensible heat fluxes. |
| Snow and Soil Energy | *messy_jsbach_sse.f90* | Soil characteristics and ground heat fluxes. |
| Turbulence | *messy_jsbach_turb.f90* | Surface roughness affecting the distribution of surface fluxes. |
| Carbon | *messy_jsbach_carb.f90* | Carbon pools above and below ground. |
| Assimilation | *messy_jsbach_assim.f90* | NPP and carbon assimilation. |
| Radiation | *messy_jsbach_rad.f90* | Surface albedo and light absorption in canopy. |
| JSBACH Lctlib | *messy_jsbach_lctlib.f90* | JSBACH land cover type (lct) library. |
| Core file (SMCL) | *messy_jsbach.f90* | JSBACH core file for MESSy. |
| Interface (SMIL) | *messy_jsbach_si.f90* | Interface for MESSy. |
| Namelist | *jsbach.nml* | User interface. |

## 2.2 The EMAC model

The ECHAM/MESSy Atmospheric Chemistry (EMAC) model is a numerical chemistry and climate simulation system that includes sub-models describing tropospheric and middle atmosphere processes and their interaction with oceans, land and human influences (Jöckel et al., 2010). It uses the second version of the Modular Earth Submodel System (MESSy2) to link multi-institutional computer codes. The core atmospheric model is the fifth-generation European Centre Hamburg general circulation model (ECHAM5, (Roeckner et al., 2006)). The physics subroutines of the original ECHAM code have been modularised and

reimplemented as MESSy submodels and have continuously been further developed. Only the spectral transform core, the flux-form semi-Lagrangian large scale advection scheme, and the nudging routines for Newtonian relaxation are remaining from ECHAM. Further details on EMAC are documented by Jöckel et al. (2016) and can be found on the MESSy website[1].

---

[1]https://www.messy-interace.org





## 2.3 Parameter optimisation

JSBACH is an alternative to the standard used submodel SURFACE. In future simulations where JSBACH is used, the SUR-
FACE submodel must be switched off in the namelist setup. Using JSBACH, a more complex scheme for land temperature
and hydrology is adopted, and with that the dynamical lower boundary conditions of ECHAM5 are modified. Since the EMAC
dynamics were optimised for the specific combination of ECHAM5 and SURFACE (Kern, 2013), the new combination of
ECHAM5 and JSBACH requires a refined parameter optimisation ("re-tuning") with respect to radiation balance, land surface
temperature, and clouds. Such a parameter optimisation is generally performed to adjust the model results as close as possible
to observations and to prevent the model climate from drifting, for example due to a large radiative imbalance. It is achieved by
small variations of specific parameters for processes with a high degree of uncertainty or a high level of parameterisation, such
as the ones related to clouds and convection. A more detailed description of the optimisation process is provided by Mauritsen
et al. (2012). The five parameters optimised in this study are the correction factor for asymmetries of ice clouds ($zasic$), the
homogeneity factors for ice and liquid water clouds ($zinhomi$ and $zinhoml$), the convective mass flux at the level of neutral
buoyancy ($cmfctop$), and the conversion factor from cloud water to rain ($cprcon$ in $s^2 m^{-2}$). Simulations for the same period
from 1990 to 2010 were carried out with gradually changed parameters. The simulation based on the default parameters is
from now on referred to as "**CTRL**". The climatically-optimized simulation is from here on referred to as "**EMAC/JSBACH**",
while the simulation without JSBACH (and with SURFACE activated instead) is referred to as "**EMAC/SRF**". The simulations
with the according parameter setups are listed in **Table S2** of the **Supplement**. Simulation 2 and 31 were not completed due to
server failures and were excluded from the analysis. Subsequently, the global and temporal average of LST, top of atmosphere
(TOA), and surface (SRF) radiation including net, solar and terrestrial parts, heat flux including net, sensible and latent parts,
total column fractional cloud coverage, total column cloud liquid and ice water content and TWS are calculated and compared
to ERA5/ERA5-Land monthly averaged data (Muñoz Sabater, 2019; Muñoz-Sabater et al., 2021). Additionally, the global and
temporal average of precipitation is compared to the GPCP monthly precipitation dataset (Adler et al., 2003). These datasets
are from here on referred to as reference data "**REF**". The results of the global and time averages of the previously mentioned
parameters for each simulation are listed in **Table S3** of the **Supplement**. The corresponding RMSE and NRMSE (RMSE nor-
malised by the range of the reference data) are shown in **Table S4** of the **Supplement**. The optimised parameters that yield the
closest fit to the reference data and with the smallest changes were selected based on the lowest normalised root mean square
error (NRMSE) sum. The sets of parameters of CTRL and EMAC/JSBACH are listed in **Table 2**. As shown here, only one
parameter needed to be adjusted via the replacement of the model level ($\mathrm{lev}$) and liquid water path ($\mathrm{lwp}$) depended calculation
of $zinhoml$ by a constant value of 0.92. The default value of $zinhoml$ is calculated based on eq. 11.52-11.53 in Roeckner
et al. (2003), viz.

$$\mathrm{zinhoml}_{\mathrm{default}} = \begin{cases} (\sum_0^{nlev} \mathrm{lwp} \partial \mathrm{lev})^{-0.1} & \text{if } (\sum_0^{nlev} \mathrm{lwp} \partial \mathrm{lev}) > 1 \\ 1 & \text{otherwise} \end{cases} \tag{1}$$



with $nlev$ being the number of model levels.

The temporally and spatially averaged results of REF, EMAC/SRF, CTRL and EMAC/JSBACH are shown in Table 3, with values that could be improved with respect to CTRL indicated in bold.

**Table 2.** List of the optimised parameters of the simulation without JSBACH (EMAC/SRF), the control simulation including JSBACH (CTRL), and the simulation best fitting the requirements (EMAC/JSBACH).

| Parameter (default) | zasic (0.85) | zinhomi (0.85) | zinhoml ($\mathrm{zinhoml_{default}}$; Eq. 1) | cmfctop (0.3) | cprcon [e-04 $s^2\,m^{-2}$] (1) |
|---|---|---|---|---|---|
| EMAC/SRF | default | default | default | default | default |
| CTRL | default | default | default | default | default |
| EMAC/JSBACH | default | default | 0.92 | default | default |

## 3    Evaluation

### 3.1    Model setup

For the present study we applied EMAC (MESSy version 2.55.0) in the T63L31MA-resolution, i.e. with a spherical truncation

of T63 (corresponding to a quadratic Gaussian grid of approx. 1.8 by 1.8 degrees spacing in latitude and longitude) with 31 vertical hybrid pressure levels up to 10 hPa. An overview of the submodels used in the reference simulation EMAC/SRF is given in **Table 4** along with a brief description of each. In the simulation EMAC/JSBACH the submodel SURFACE is replaced by the new submodel JSBACH, and the tuning parameters are updated, while the remaining set-up is unchanged. Both simulations were performed from January 1970 to January 2011 and include tracer nudging of $CO_2$, $CH_4$, $N_2O$, CFC11 and CFC12 based

on tracer profiles derived from Atmospheric Chemistry and Climate Model Intercomparison Project (ACCMIP) historical lower boundary condition datasets (Lamarque et al., 2010, 2013). JSBACH is operated on five snow and three canopy layers and five soil layers, reaching a depth of 9.8 meter below the surface. From 21 possible land cover types (LCTs), 11 plant functional types (PFTs) are taken into account for the standard 11 tile setup (**Appendix Table A1**). Those are tropical and extra-tropical broadleaf evergreen and deciduous trees, rain-green shrubs, deciduous shrubs, $C_3$- and $C_4$-grass, $C_3$- and $C_4$-pasture, $C_3$-

and $C_4$-crops. This evaluation focuses only on the dynamical coupling between EMAC and JSBACH, thus all calculations of atmospheric chemistry and the O-GCM are deactivated. A list of all coupled variables can be found in the namelist attached in the electronic **Supplement**. Coupled variables include surface temperature, latent and sensible fluxes, ground heat fluxes, soil water content, surface albedo and specific humidity at lowest atmospheric level. Atmospheric Model Intercomparison Project (AMIP) type simulations were performed with prescribed monthly sea surface temperature and sea ice concentration to identify

systematic errors in the model (Gates et al., 1999). The sea surface temperature and ice concentration is derived from ERA5 six hourly data from 1940 to present (Hersbach et al., 2020). JSBACH was initialized with carbon pools, soil and land property data from the year 2005 (**Supplement**), which are estimated to stabilize within five years. This is possible since we perform



**Table 3.** Table of the temporally and globally averaged results $\pm$ inter-annual variability as the standard deviation of CTRL, EMAC/SRF and EMAC/JSBACH from 1990 to 2010. The corresponding reanalysis or observational results are listed as "REF". For precipitation REF refers to the GPCP monthly precipitation dataset (Adler et al. (2003)), while for the remaining variables, REF refers to ERA5/ERA5-Land reanalysis datasets (Hersbach (2023); Muñoz Sabater(2019,2021)).

| Run | LST | $TOA_{net}$ | $TOA_{sw}$ | $TOA_{lw}$ | $SRF_{net}$ |
| --- | --- | --- | --- | --- | --- |
| | [K] | [W m$^{-2}$] | [W m$^{-2}$] | [W m$^{-2}$] | [W m$^{-2}$] |
| REF | 282.25 $\pm$ 0.27 | 0.45 $\pm$ 0.65 | 242.67 $\pm$ 0.65 | -242.22 $\pm$ 0.29 | 105.91 $\pm$ 0.45 |
| EMAC/SRF | 283.09 $\pm$ 0.27 | 3.56 $\pm$ 0.39 | 234.33 $\pm$ 0.27 | -230.77 $\pm$ 0.34 | 107.92 $\pm$ 0.24 |
| CTRL | 280.71 $\pm$ 0.26 | 7.41 $\pm$ 0.47 | 237.46 $\pm$ 0.37 | -230.06 $\pm$ 0.39 | 108.52 $\pm$ 0.25 |
| EMAC/JSBACH | 280.48 $\pm$ 0.23 | 3.23 $\pm$ 0.38 | 233.86 $\pm$ 0.29 | -230.63 $\pm$ 0.38 | 104.43 $\pm$ 0.3 |

| Run | $SRF_{sw}$ | $SRF_{lw}$ | $HFLX_{net}$ | $HFLX_{sensible}$ | $HFLX_{latent}$ |
| --- | --- | --- | --- | --- | --- |
| | [W m$^{-2}$] | [W m$^{-2}$] | [W m$^{-2}$] | [W m$^{-2}$] | [W m$^{-2}$] |
| REF | 163.76 $\pm$ 0.54 | -57.85 $\pm$ 0.31 | -69.92 $\pm$ 0.57 | -28.15 $\pm$ 0.68 | -41.76 $\pm$ 0.43 |
| EMAC/SRF | 161.74 $\pm$ 0.31 | -53.83 $\pm$ 0.3 | -104.24 $\pm$ 0.35 | -16.74 $\pm$ 0.18 | -87.5 $\pm$ 0.42 |
| CTRL | 166.09 $\pm$ 0.45 | -57.58 $\pm$ 0.4 | -110.68 $\pm$ 0.43 | -11.68 $\pm$ 0.11 | -99.0 $\pm$ 0.45 |
| EMAC/JSBACH | 162.14 $\pm$ 0.34 | -57.71 $\pm$ 0.35 | -110.47 $\pm$ 0.67 | -11.67 $\pm$ 0.14 | -98.79 $\pm$ 0.61 |

| Run | Precip | ACLC | LWC | IWC | TWS |
| --- | --- | --- | --- | --- | --- |
| | [mm day$^{-1}$] | | [kg m$^{-2}$] | [kg m$^{-2}$] | [m] |
| REF | 2.7 $\pm$ 0.03 | 0.553 $\pm$ 0.00405 | 0.04707 $\pm$ 0.00098 | 0.02166 $\pm$ 0.00033 | 1.06012 $\pm$ 0.00947 |
| EMAC/SRF | 2.83 $\pm$ 0.02 | 1.06067 $\pm$ 0.00444 | 0.10394 $\pm$ 0.00115 | 0.04972 $\pm$ 0.00068 | 0.34995 $\pm$ 0.00425 |
| CTRL | 2.75 $\pm$ 0.02 | 0.6452 $\pm$ 0.0032 | 0.09538 $\pm$ 0.00114 | 0.04943 $\pm$ 0.00068 | 1.00555 $\pm$ 0.00291 |
| EMAC/JSBACH | 2.76 $\pm$ 0.02 | 0.6464 $\pm$ 0.0028 | 0.09519 $\pm$ 0.0009 | 0.04936 $\pm$ 0.00054 | 1.00385 $\pm$ 0.00815 |

AMIP type simulations in which the land-carbon interaction remains inactive. Atmospheric variables stabilize within days and the soil moisture is estimated to be the slowest variable to adjust to equilibrium, with a maximum adjustment time of one year (Hagemann and Stacke, 2015; Schneck et al., 2022). Therefore the first year (1970) is considered as spin-up time and is not taken into account for the evaluation.



**Table 4.** List of the submodels comprising the EMAC/SRF simulation including short description and reference.

| PROCESS SUBMODELS | Short Description | Reference |
|---|---|---|
| AEROPT | Calculation of aerosol optical properties. | Dietmüller et al. (2016) |
| CLOUD | ECHAM5 cloud scheme as MESSy submodel. | Roeckner et al. (2006); Tost |
| CLOUDOPT | Calculation of cloud optical properties. | Dietmüller et al. (2016) |
| CONVECT | Convection parametrisations. | Tost et al. (2006) |
| E5VDIFF | Land-atmosphere exchange and vertical diffusion based on ECHAM5. | MESSy; Roeckner et al. (2003) |
| GWAVE | ECHAM5 non-orographic gravity wave routines plus additional drag parametrisiations. | MESSy; Hines (1997) |
| HD | Hydrological Discharge model for present day rivers. | Pozzer et al. (2011a) |
| TNUDGE | Newtonian relaxation of species as pseudo-emission. | Kerkweg et al. (2006) |
| ORBIT | Calculation of orbital parameters of the Earth orbit around the sun. | Dietmüller et al. (2016) |
| OROGW | Parameterisation of drag due to subgrid scale orography blocking and orographic gravity wave forcing. | Chapter 7 of Roeckner et al. (2003) |
| PTRAC | User-defined initialised Prognostic Tracers. | Jöckel et al. (2008) |
| RAD | ECHAM5 radiation code with extended features. | Dietmüller et al. (2016) |
| SURFACE | Modularized version of the ECHAM5 subroutines SURF, LAKE, LICETEMP and SICETEMP. | Chapter 6 of Roeckner et al. (2003) |

### 3.1.1 Evaluation variables and reference datasets

The selected variables to be assessed are variables representing the land surface like Land Surface Temperature (LST), Terrestrial Water Storage (TWS) and Surface Albedo ($\alpha$), but also other atmospheric variables like precipitation (precip), top of atmosphere radiation balance ($Rad_{TOA}$), Fraction of Absorbed Photosynthetic Active Radiation (FaPAR), Leaf Area Index (LAI), and Gross Primary Productivity (GPP) in line with the study of Schneck et al. (2022). These variables are either compared to ERA5/ERA5-LAND reanalysis data-sets, or directly to observational data-sets of the GPCP or MODIS satellite data (**Table 5**). The ERA5/ERA-Land reanalysis data is a combination of synthesized estimates of the climate state, which are calculated based on as many observations as possible, and a numerical model either due to direct assimilation of the observations or due to forcings (Muñoz-Sabater et al., 2021).





**Table 5.** Table of the evaluation Variables, corresponding reference dataset and the analysed time period of the evaluation.

| Variable | Dataset | Time period | Reference |
|---|---|---|---|
| LST | | | |
| TWS | ERA5-Land monthly averaged data 1950 to present | 1971 - 2010 | Muñoz Sabater (2019, 2021) |
| Surface Albedo | | | |
| Rad$_{TOA}$ | ERA5 monthly averaged data 1940 to present | 1971 - 2010 | Hersbach (2023) |
| Precipitation | GPCP monthly precipitation dataset 1979 to 2021 | 1980 - 2010 | Adler et al. (2003) |
| LAI faPAR | MOD15A2H MODIS/Terra Leaf Area Index/FaPAR 8-Day L4 Global 500m SIN Grid V061 regridded to globally data at 0.5 resolution derived by ICDC | 2000 - 2010 | Kern (2021); Running et al. (2015) |
| GPP | MOD17A2H MODIS/Terra Gross Primary Productivity 8-Day L4 Global 500m SIN Grid V006 regridded to globally data at 0.5 resolution derived by ICDC | 2000 - 2010 | Kern (2021); Running et al. (2015) |

# 4 Results and Discussion

To get an overview of the model performance compared to the reference data, their statistical metrics of the monthly and globally averaged results are presented in **Fig. 1** as a Taylor diagram (Taylor, 2001). For the classification of the results of the coupled model, the statistical measures of the model results without the coupling to JSBACH are added. Results of the EMAC/JS-

BACH are shown as dots, while the results of the EMAC/SRF simulation are displayed as crosses. The Taylor plot shows the Pearson correlation coefficient between simulated and reference data on straight lines from the origin, with arcs around the origin indicating the standard deviation normalised by the reference standard deviation, and arcs around the value 1 indicating the root mean square error normalised by the range of the reference data (NRMSE). The Pearson correlation coefficient (r) and NRMSE are listed in **Table 6**, together with the weighted global average ± standard deviations for the model simulations

($\overline{MOD}$) and reference data ($\overline{REF}$). The correlation and NRMSE are based on monthly averages for the available time period of the reference datasets, with the correlation conducted over time and location (see Fig. 5). This covers the years 1971-2010 for LST, Rad$_{TOA}$, $\alpha$ and TWS. Precipitation is analysed for the period 1980-2010 and LAI, FaPAR and GPP for 2000-2010. LST (shown in red in **Fig. 1**) has the highest correlation with REF, 0.985 for EMAC/JSBACH and 0.989 for EMAC/SRF. Whereas the global average EMAC/JSBACH LST is on average $1.546\,\text{K}$ colder than REF and EMAC/SRF $0.743\,\text{K}$ warmer than REF.

The second highest correlation between REF and the model simulations is found for the surface albedo (shown in black in **Fig. 1**), 0.946 for EMAC/JSBACH and 0.943 for EMAC/SRF. Also for this parameter, EMAC/JSBACH has a slightly lower global average than REF with an average difference of $-0.015$ and EMAC/SRF differ by $-0.012$ from the global average. The third highest correlation is found for Rad$_{TOA}$ (shown in orange in **Fig. 1**), 0.907 for EMAC/JSBACH vs. REF and 0.909 for EMAC/SRF vs. REF. The net radiative flux at the top of the atmosphere (TOA) between EMAC/JSBACH and REF differs by



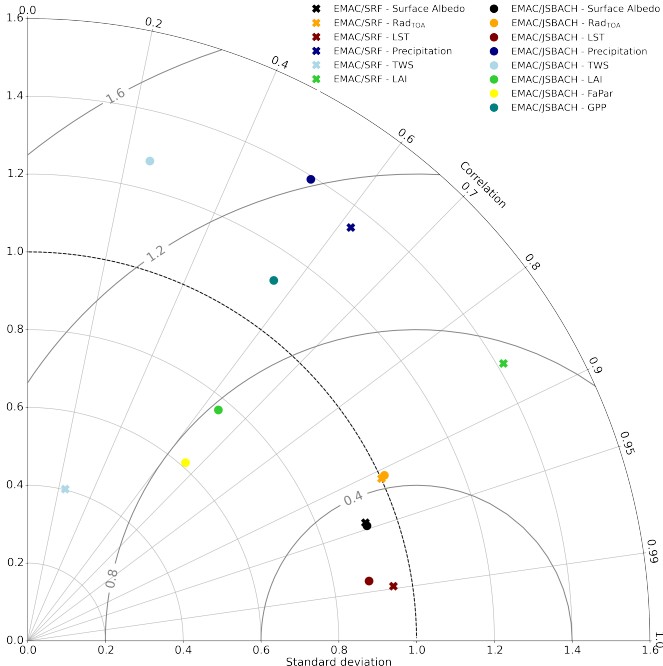

**Figure 1.** Taylor plots of the Pearson correlation coefficient, root mean square error (RMSE), and standard deviation normalised with the reference data standard deviation of monthly means of LST (1971-2010), Rad$_{\text{TOA}}$ (1971-2010), surface albedo (1971-2010), precipitation (1980-2010), TWS (1971-2010), LAI (2000-2010), faPAR (2000-2010) and GPP (2000-2010). The statistical measures of the EMAC/JS-BACH are displayed as dots, while the EMAC/SRF values are shown as crosses. As EMAC does not provide faPAR and GPP output, both are only shown for the EMAC/JSBACH simulation. Straight lines from the origin represent the correlations, while the arcs around the origin represent the standard deviations. The arcs around the value 1 on the horizontal axis show the RMSE normalised to the standard deviation. The correlation and RMSE are based on monthly average values, with the correlation conducted over time and location.

$3.56\,\text{Wm}-2$ and $3.045\,\text{Wm}-2$ EMAC/SRF. FaPAR (shown in yellow in **Fig. 1**) is only available for the EMAC/JSBACH simulation with a correlation of $0.663$ with REF. The global average difference is $0.223$. The correlation of simulated and observed LAI (shown green in **Fig. 1**) is $0.637$ for the EMAC/JSBACH simulation and $0.864$ for the climatology used in EMAC/SRF. EMAC/JSBACH underestimates the global vegetation LAI by $0.212$, while the EMAC/SRF climatology overestimates LAI by $0.768$. As for faPAR, GPP (shown dark green in **Fig. 1**) is only available for EMAC/JSBACH leading to a correlation with

REF of $0.564$ with an average global difference to REF of $-0.001\,\text{kg}(\text{Carbon})\,\text{km}^{-2}$. The correlation between simulated precipitation (shown in dark blue in **Fig. 1**) and REF is $0.523$ for EMAC/JSBACH and $0.614$ for EMAC/SRF. EMAC/JSBACH overestimates the global mean precipitation by $0.042\,\text{mmday}^{-1}$ and EMAC/SRF by $0.316\,\text{mmday}^{-1}$. The lowest correlation between model results and reference data is found for TWS (shown in light blue in **Fig. 1**). EMAC/JSBACH and REF correlate with $0.251$ while EMAC/SRF and REF correlate with $0.242$. EMAC/JSBACH overestimates the mean global TWS by $0.052\,\text{m}$,

and EMAC/SRF underestimates it by $0.68\,\text{m}$.




For a more comprehensive assessment, in the following subsections each variable derived from the new coupled model is evaluated individually via the comparison to the reference dataset and the EMAC/SRF simulation. Monthly averages over the corresponding analysis period were calculated and subsequently, the average values for the spring and summer months (March, April, May, June, July, and August) and the autumn and winter months (September, October, November, December, January, and February) were determined. The same was done for the MODIS, GPCP and the ERA5 datasets.

**Table 6.** Summary of the comparison between model results and reference data. The Pearson correlation coefficient is listed as r, NRMSE shows the root mean squared error normalised by the range of the reference data. The simulations were performed at T63L31 resolution and with a model time step of 600s. The columns $\overline{\text{MOD}}$ and $\overline{\text{REF}}$ are the weighted global averages ± standard deviation of the model simulation results and the reference data.

| | Variable | Period | r | NRMSE | $\overline{\text{MOD}}$ | $\overline{\text{OBS}}$ |
|---|---|---|---|---|---|---|
| *EMAC/JSBACH vs. OBS* | | | | | | |
| | LST [K] | 1971-2010 | 0.985 | 0.045 | 280.434 ±25.09 | 281.98 ±28.15 |
| | TWS [m] | 1971-2010 | 0.247 | 0.26 | 1.13 ±0.706 | 1.078 ±0.56 |
| | Surface Albedo | 1971-2010 | 0.947 | 0.127 | 0.301 ±0.285 | 0.316 ±0.309 |
| | $\text{Rad}_{\text{TOA}}$ [W m$^{-2}$] | 1971-2010 | 0.907 | 0.099 | 3.948 ±95.776 | 0.388 ±94.729 |
| | Precipitation [mm day$^{-1}$] | 1980-2010 | 0.523 | 0.083 | 2.738 ±3.382 | 2.696 ±2.43 |
| | LAI | 2000-2010 | 0.637 | 0.175 | 1.187 ±1.11 | 1.399 ±1.44 |
| | faPAR | 2000-2010 | 0.663 | 0.271 | 0.161 ±0.16 | 0.384 ±0.25 |
| | GPP [kg(Carbon) km$^{-2}$] | 2000-2010 | 0.564 | 0.203 | 0.02 ±0.02 | 0.021 ±0.02 |
| *EMAC/SRF vs. OBS* | | | | | | |
| | LST [K] | 1971-2010 | 0.989 | 0.037 | 282.796 ±26.772 | |
| | TWS [m] | 1971-2010 | 0.241 | 0.308 | 0.394 ±0.223 | |
| | Surface Albedo | 1971-2010 | 0.944 | 0.129 | 0.303 ±0.285 | |
| | $\text{Rad}_{\text{TOA}}$ [W m$^{-2}$] | 1971-2010 | 0.909 | 0.098 | 3.434 ±94.794 | s.a. |
| | Precipitation [mm day$^{-1}$] | 1980-2010 | 0.616 | 0.074 | 3.025 ±3.279 | |
| | LAI (climatology) | 2000-2010 | 0.864 | 0.263 | 2.165 ±2.038 | |

## 4.1 Land Surface Temperature (LST)

The land surface temperature is one of the main drivers in determining the habitat conditions for the vegetation and living organisms in the ecosystems. It is one of the most important drivers of all land processes, as it controls the surface energy and radiation balance, as well as the hydrological and thermal exchange fluxes between the surface and the atmosphere. Furthermore, it determines the freezing and thawing of the snow and ice covers. It is the upper boundary condition for the soil temperature calculation within the five layer soil scheme and one of the lower boundary conditions for EMAC. LST is calculated in JSBACH via the surface energy balance equation and the values for saturated humidity and dry static energy,





based on the Richtmyer-Morton coefficients derived from the vertical diffusion scheme of ECHAM (Reick et al., 2021). Here, LST is compared with monthly averaged reanalysis data from ERA5-Land available from 1950 to the present (Muñoz Sabater,

2019, 2021), with only the simulated years (1971-2010) included for the comparison. The ERA5-Land dataset is provided at $0.1°x0.1°$ spatial resolution and is interpolated to the T63L31 EMAC output grid.

The geographical distribution of the LST difference between ERA5 and EMAC/JSBACH is shown in **Fig. 2**. **Figure 3** and **Fig. 4** show the LST trend and seasonality analysis and **Fig. 5** shows the LST seasonality and corresponding latent heat flux at the surface separated for three climate zones. The polar zone is defined as latitudes $> 66.5°$, the temperate zone as latitudes

between $40°$ and $66.5°$ and the tropical and subtropical zone as latitudes $< 40°$. As shown in **Fig. 2**, the EMAC/JSBACH LST is lower everywhere throughout the year, except for the polar regions, the Himalaya and over the Amazon basin. The largest LST underestimations are found over the Rocky Mountains and the Taklamakan- and Gobi deserts, being most pronounced in the Northern Hemispheric summer. The largest overestimation of LST occurs over the Antarctic Ross Ice Shelf (up to $20\,K$), along the coast of the Greenland Sea (up to $15\,K$) and the Hindu Kush, Himalayan, Kuen Lun and Tien Shen mountain

ranges (up to $15\,\mathrm{K}$). As result, the zonal mean shows a slightly warmer surface in the polar regions and a colder surface in the subtropics and temperate zones. In the temporal and global average, the LST of EMAC/JSBACH is $1.546\,K$ colder compared to the reanalysis data. The general trend of steadily increasing LST, is reproduced by EMAC/JSBACH (**Fig. 3**). Nevertheless, in both - the trend analysis and seasonality analysis - the overall difference of $1.546\,K$ between EMAC/JSBACH an ERA5 LST is clearly visible. When comparing the geographical difference between EMAC/SRF and ERA5 LST, as displayed in **Fig. 2**,

overestimations of the LST are found over the Antarctic Ross Ice Shelf, along the coast of the Greenland Sea, and over the same mountain ranges as previously found for the EMAC/JSBACH results. Here too, the polar regions are in general warmer than indicated by ERA5. Overall, the LST derived from EMAC/SRF is $0.743\,K$ warmer than the reanalysis data. This is also visible in the trend analysis, which shows the overall warmer land surface of EMAC/SRF in comparison to ERA5. In the zonal mean, the differences largely cancel out, leading to a similar zonal progression of the EMAC/SRF results compared to the

ERA5 LST.

Comparing EMAC/JSBACH and the ERA5 LST, larger differences than $1.546\,K$ are found for the tropics, subtropics and temperate zone (latitudes $> 66.5°$), while in the polar climate zone the LST is overestimated in both simulations as shown in **Fig. 5**. The lower LST of EMAC/JSBACH compared to EMAC/SRF in non-polar regions can be explained by variations of the latent heat flux, where EMAC/JSBACH simulated consistently higher values than EMAC/SRF. Here, the main driver is

evapotranspiration, the process by which water vapour is released from the surface and vegetation. Evapotranspiration has a cooling effect on the surface, due to the energy absorbed during the phase change of the water. As a result, cooler LST values are found in regions where evapotranspiration is more intense, such as the tropics and extra-tropics. In the polar regions, where vegetation is sparse or absent, the difference in latent heat flux between EMAC/JSBACH and EMAC/SRF is less significant, as shown in **Fig. 5**, resulting in less variation of the LST between simulations. The strong overestimation of LST along the

Antarctic coast is visible in both simulations and might be an artifact of sea ice occurrence and the variability of snow and ice cover. Despite local variations, the overall temporal and spatial correlation between EMAC/JSBACH and ERA5 is large (0.985,



**Table 6**) indicating that LST is in general realistically reproduced, and the representation of seasonal patterns and overall trends are plausible.

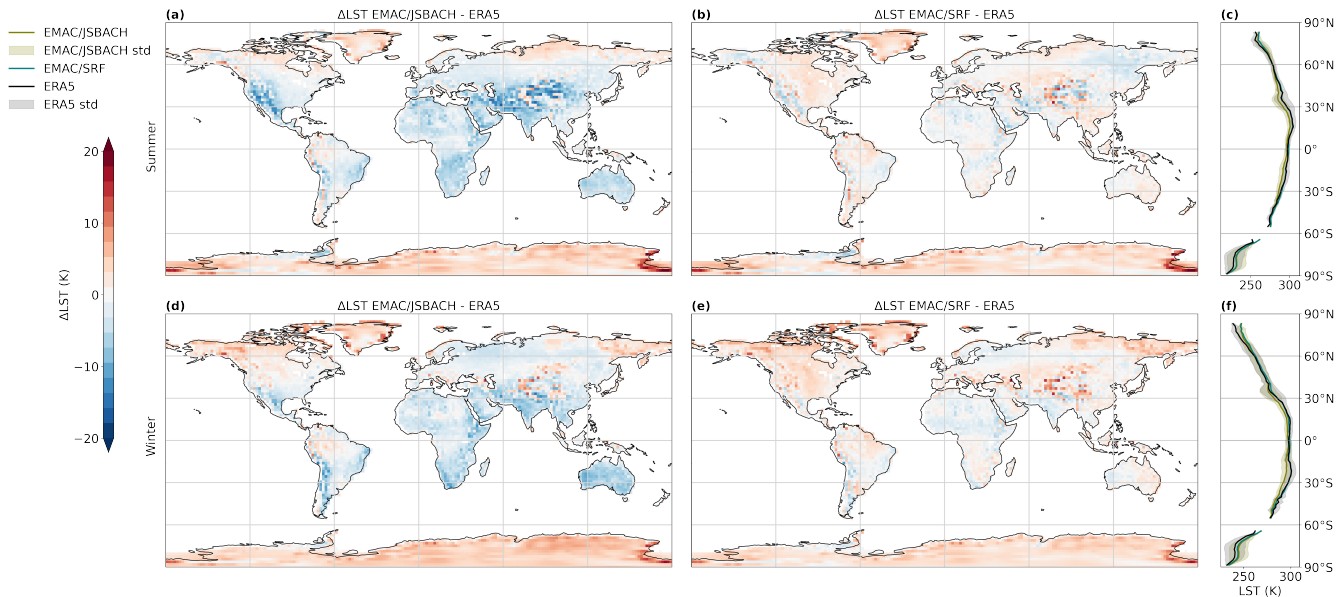

**Figure 2.** Difference of Land Surface Temperature (LST) between EMAC/JSBACH and ERA5-Land during Northern Hemispheric (NH) summer (a) and NH winter (d) months, with data averaged over the years 1971 to 2010. Analogously the difference between EMAC/SRF and ERA5-Land LST during summer (b) and winter (e) months is displayed. Positive values represent an overestimation of the simulated LST, while negative values indicate an underestimation. Additionally, the zonal average of all three datasets for both summer (c) and winter months (f) is shown. Here, LST from EMAC/JSBACH is depicted in green, LST from EMAC/SRF is shown in blue, and LST from the ERA5-Land dataset is represented in black. The shaded area within the zonal mean plot illustrates the standard deviations along longitudes.





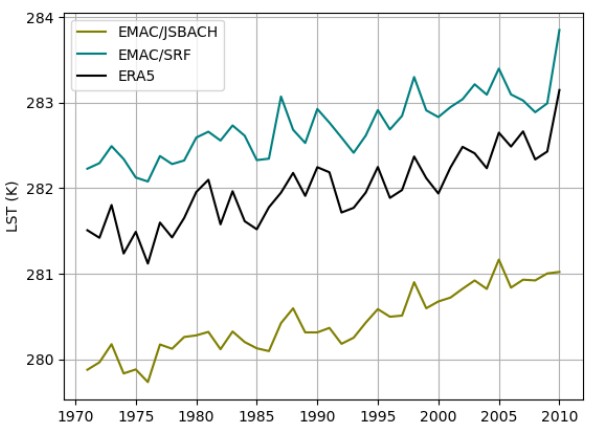

**Figure 3.** Globally averaged LST trend (in $K$) of EMAC/JSBACH (green), EMAC/SRF (blue) and ERA5 (black).

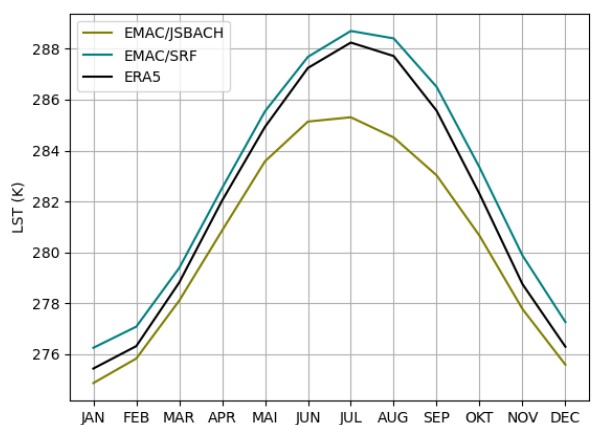

**Figure 4.** Globally averaged seasonal LST (in $K$) of EMAC/JSBACH (green), EMAC/SRF (blue) and ERA5 (black) for the years 1971 to 2010.

## 4.2 Terrestrial Water Storage (TWS)

TWS is defined as the vertical integrated water content on land and the subsurface including groundwater, rivers, lake water, soil moisture (also in the root zone), snow and ice (including permafrost), wet biomass, and water stored in vegetation (Girotto and Rodell, 2019). It depends on the amount of precipitation and the air temperature as well as on the soil type and infiltration, vegetation cover, surface and soil temperature and runoff (Schneck et al., 2022). In EMAC/JSBACH the TWS is the sum of water content and runoff, calculated separately. The water content is calculated as the sum of all water reservoirs above and

below ground, down to the bedrock. Everything below the bedrock, like deep ground water and aquifers, are not taken into account (Reick et al., 2021). The above ground water includes the wet skin reservoir (water on canopy and surface) and snow on canopy and surface, both depending on - and exchanging moisture between surface and atmosphere via - precipitation, sublimation, melting and wind-blow. The surface runoff is accounted for as the water reaching the surface from above, but not infiltrating the soil. Water infiltrating the ground either percolates by gravitational movement (ending up as drainage, if it

reaches bedrock) or diffuses. Depending on its phase, it is defined as one of the EMAC/JSBACH below ground water reservoirs: soil moisture or soil ice. At the surface, the moisture exchange to the atmosphere occurs through evapotranspiration, dew formation, or evaporation of bare soil, controlled by the specific humidity and temperature of the surrounding air. Furthermore, these processes strongly depend on vegetation coverage and, with that, plant productivity, which will also be assessed via the gross primary productivity (GPP) (Section 4.8). TWS is compared to the ERA5-Land TWS, derived as the sum of the integrated

volumetric soil water content, skin reservoir content and runoff. The ERA5 soil water column is distributed over 4 layers, with





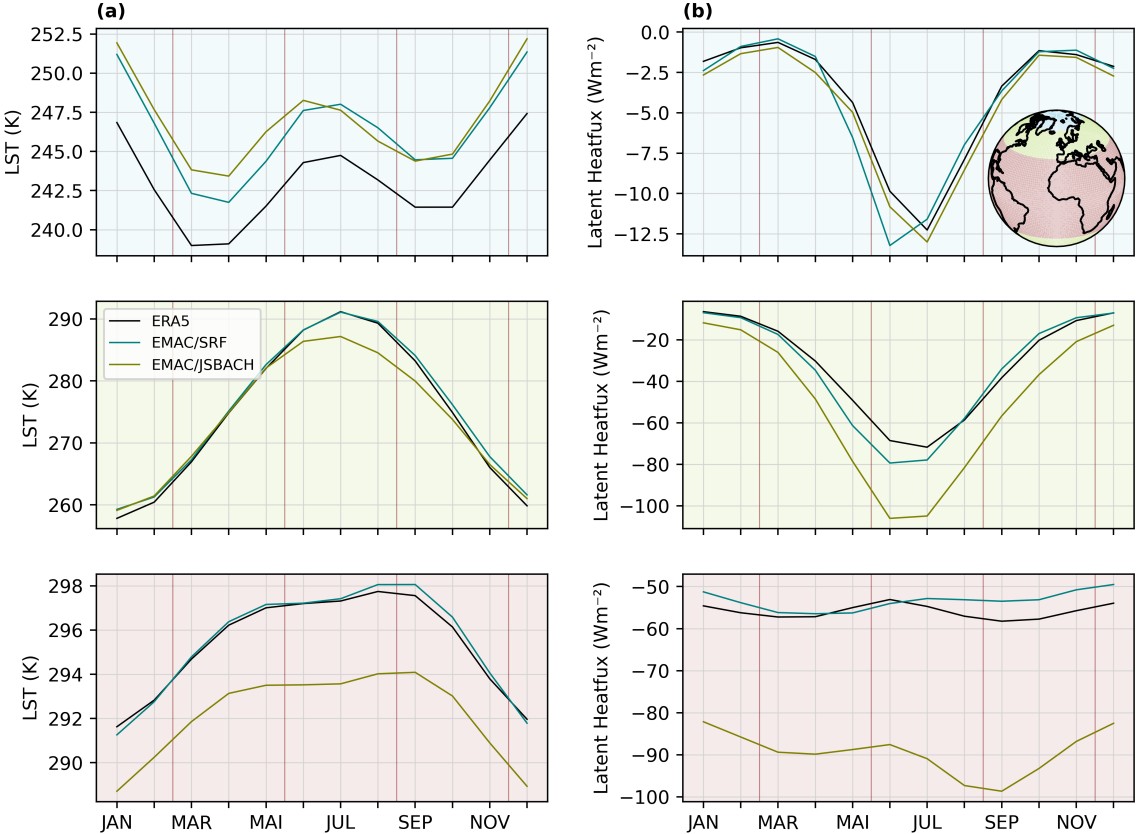

**Figure 5.** Averaged seasonal LST (in $K$), surface latent heatflux (in $W\,m^{-2}$), low cloud cover (lcc), medium cloud clover (mcc) and high cloud cover (hcc) of EMAC/JSBACH (green), EMAC/SRF (blue), ERA5 (black) for the years 1971 to 2010. The blue background colour indicates values averaged over the polar climate zone (latitudes $> 66.5\,^\circ$), the green background colour indicates values averaged over the temperate climate zone (latitudes between $40\,^\circ$ and $66.5\,^\circ$), and the red background colour indicates values averaged over the tropical and subtropical climate zone (latitudes $< 40\,^\circ$).

a maximum depth of 2.89 meters. Here too, the ERA5 dataset is interpolated to the EMAC grid. Since TWS is not calculated for glaciers within EMAC/JSBACH, glaciated polar areas are excluded from this analysis.

In **Fig. 7** the difference of TWS between EMAC/JSBACH and ERA5 is shown. The annual global average of EMAC/JS-BACH TWS weighted by latitudes is $0.052\,m$ larger than the global mean of the ERA5. The maximum overestimation of TWS is found in Russia (up to $3.0\,m$). EMAC/JSBACH overestimates TWS almost everywhere, except for high elevated regions



such as the Tibetan Plateau and Tien Shen, central and eastern Siberia, India (Deccan Plateau), the Ethiopian highlands and Patagonia. Additionally, TWS is underestimated in the Amazon Basin.

The zonal mean of the EMAC/JSBACH TWS, as shown in **Fig. 7**, does not exactly reproduce the zonal mean of the TWS from ERA5, but its absolute values are in better agreement with the ERA5 data than the EMAC/SRF results, which is also 300 visible in **Fig. 6**. The TWS of the EMAC/SRF simulation is lower everywhere compared to the ERA5 data-set, except for deserts, where the TWS is anyhow low. This leads to an annual global average of the EMAC/SRF TWS of $0.394 \pm 0.223\,m$, $-0.68\,m$ lower than the one derived from reanalysis data.

The soil hydrology module that comes with EMAC/JSBACH offers the possibility to improve the representation of the soil water. the soil moisture in EMAC/SRF was computed based on a simple bucket model. Following Seneviratne et al. (2010), this 305 is now replaced by a more complex five-layer diffusive hydrological transport model that includes water storage and infiltration in five soil layers, preventing too rapid soil drying. While EMAC/SRF tends to strongly underestimate soil moisture levels everywhere, the integration of JSBACH results in larger and more spatially diverse soil moisture content. However, de Vrese et al. (2022) found that in the JSBACH version used here, infiltration only takes place if the temperature of the first soil layer is at or above the melting point. In combination with the five-layer snow scheme presented by Ekici et al. (2014), this becomes 310 problematic. During spring snowmelt, soil temperatures are below the 0 ∘C of the overlying snow cover, causing all the melt water to run off at the surface, while in reality a considerable amount should percolate into the soil (de Vrese et al., 2022). This contributes to the strong underestimation of TWS in permafrost regions, e.g. Siberia. Despite this, the global average trend analyses shows that EMAC/JSBACH TWS aligns much more closely with observed TWS compared to the EMAC/SRF results (**Fig. 6**).

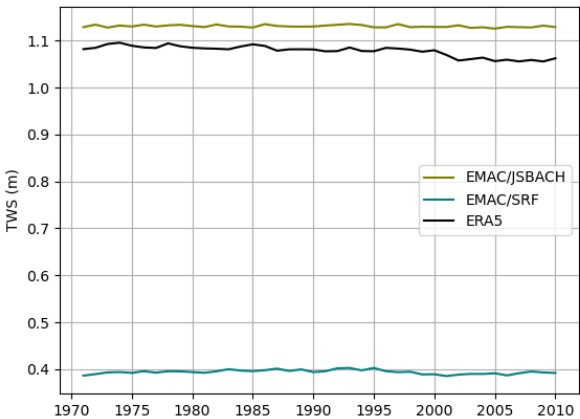

**Figure 6.** Globally averaged TWS trend (in $kgkg^{-1}$) of EMAC/JSBACH (green), EMAC/SRF (blue) and ERA5 (black).



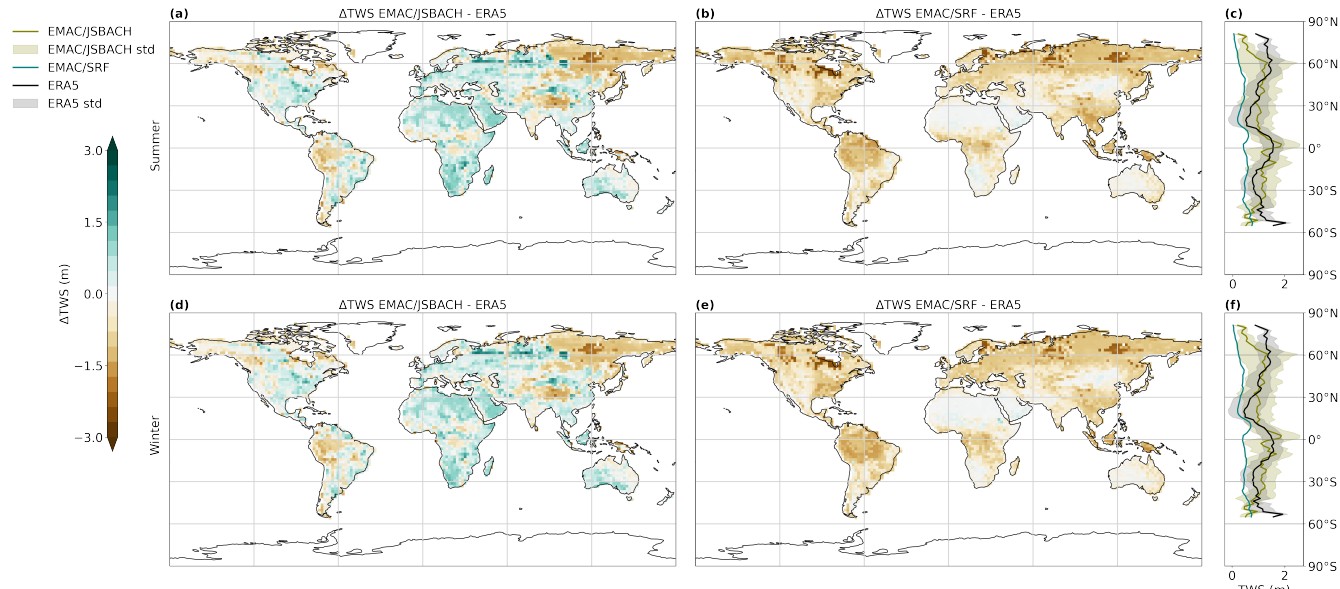

**Figure 7.** Difference of Terrestrial Water Storage (TWS) between EMAC/JSBACH and ERA5-Land during NH summer (a) and NH winter (d) months, with data averaged over the years 1971 to 2010. Analogously the difference between EMAC/SRF and ERA5-Land TWS during summer (b) and winter (e) months is displayed. Positive values represent an overestimation of the simulated TWS, while negative values indicate an underestimation. Additionally, the zonal average of all three datasets for both summer (c) and winter months (f) is shown. Here, TWS from EMAC/JSBACH is depicted in green, TWS from EMAC/SRF is shown in blue, and TWS from the ERA5-Land dataset is represented in black. The shaded areas within the zonal mean plots illustrate the standard deviations of the datasets. Glaciated polar regions are excluded.



### 4.3 Surface Albedo ($\alpha$)

Another key factor in Earth system modelling is the surface albedo, as it is a fundamental input for the radiation scheme and strongly influences the energy budget of the planet. Generally defined as the reflected fraction of incoming solar radiation, it depends on the type of land cover and the extent and thickness of the snow cover or ice sheet. Especially over the continental area of the Northern Hemisphere and the sea ice cover in the Southern Hemisphere, the surface albedo can exert a strong feedback effect (Hall, 2004). Since the surface coverage of snow and ice can vary on small scales and is strongly coupled to atmospheric and oceanic dynamics, the computation of the surface albedo is still a challenging factor for GCMs (Bony et al., 2006). In EMAC/JSBACH, the surface albedo on glaciers is calculated either for grid boxes with ice sheets or without - where these boxes are either completely or not at all covered with ice. For ice sheets, the albedo is calculated according to ECHAM5 (Roeckner et al., 2003). Every other surface is treated with a new albedo scheme based on the current state of snow cover, LAI, vegetation distribution, and the spectral composition of solar downward radiation for each grid box, as described by Reick et al. (2021). The surface albedo is compared to the ERA5-Land albedo variable. The ERA5-Land albedo is based on a 5-year MODIS climatology. These satellite observations are a combined Terra and Aqua retrieval (?Schaaf and Wang, 2015). From the 16-day level-3 data of a $0.05^\circ$ climate modelling grid (approx. 5.6 km at the equator) monthly averages are calculated and interpolated to the EMAC T63 grid. The surface albedo of EMAC/JSBACH is in general in good agreement with the ERA5 surface albedo (**Fig. 8**). During summer, EMAC/JSBACH shows a slight overestimation in the Northern Hemisphere and an underestimation in the Southern Hemisphere. In the winter months the opposite applies, underestimation in the Northern Hemisphere and overestimation in the Southern Hemisphere. The average annual global difference between EMAC/JSBACH and ERA5 is $-0.015$. The same geographical patterns are visible for the EMAC/SRF vs. ERA5 surface albedo comparison with average annual global difference of $-0.012$. The zonal mean shows a slight overestimation of surface albedo for both simulations in the southern subtropics during summer and winter. During winter there is a minimum of the surface albedo at about $45^\circ$ N, which is not seen in the reference data (**Fig. 8**).

The land surface albedo remains almost unchanged in the new model version. This is presumably due to the fact that in EMAC/SRF the background albedo is temporally constant except for changes in ice and snow cover Nützel et al. (2023). In EMAC/JSBACH a simplified ground albedo scheme was used to obtain a comparable results. However, there is a slight improvement compared to the reference data for EMAC/JSBACH, particularly noticeable during the summer in eastern Siberia, where the model overestimation is reduced. In this region, the LST and LAI derived from EMAC/JSBACH align more closely to the reference data than EMAC/SRF alone. This suggests that the slightly warmer surface and less vegetation in this region may contribute to the improved surface albedo representation.



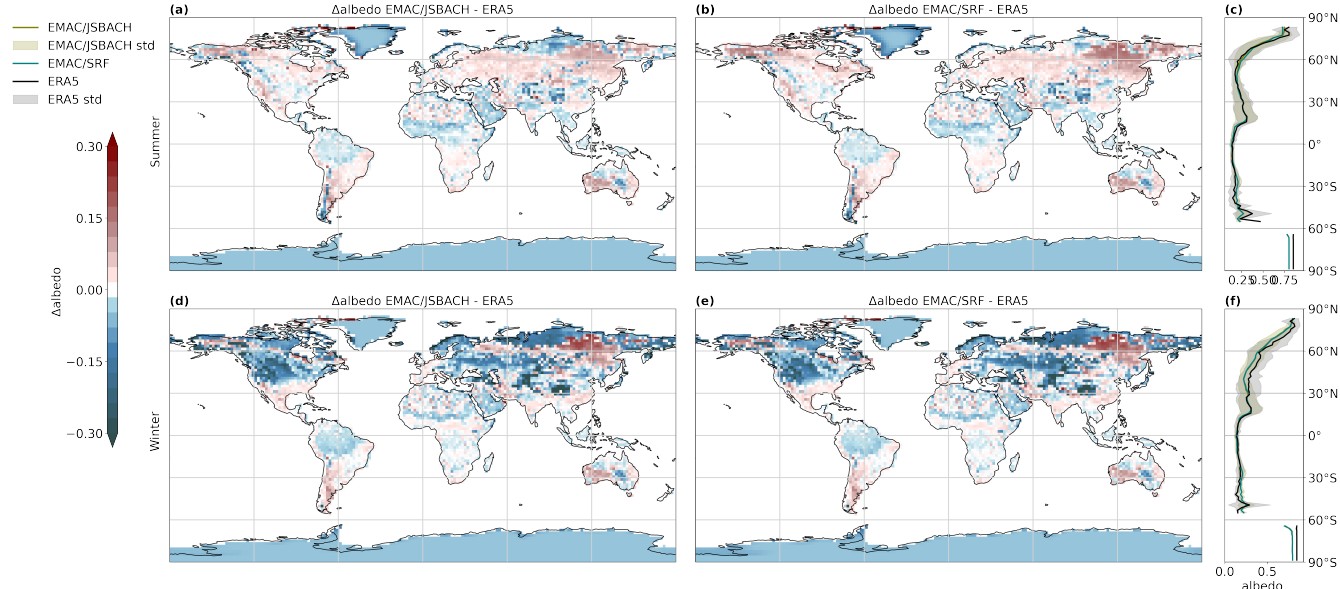

**Figure 8.** Difference of surface albedo ($\alpha$) between EMAC/JSBACH and ERA5-Land during Northern Hemispheric (NH) summer (a) and NH winter (d) months, with data averaged over the years 1971 to 2010. Analogously the difference between EMAC/SRF and ERA5-Land $\alpha$ during summer (b) and winter (e) months is displayed. Positive values represent an overestimation of the simulated $\alpha$, while negative values indicate an underestimation. Additionally, the zonal average of all three datasets for both summer (c) and winter months (f) is shown. Here, $\alpha$ from EMAC/JSBACH is depicted in green, $\alpha$ from EMAC/SRF is shown in blue, and $\alpha$ from the ERA5-Land dataset is represented in black. The shaded area within the zonal mean plot illustrates the standard deviations along longitudes.



## 4.4 Top of atmosphere radiation balance Rad$_{\mathrm{TOA}}$

The top of atmosphere (TOA) net radiation can be defined as the difference between the incoming solar radiation, outgoing solar radiation back-scattered and reflected by clouds, aerosols, air and the land surface, and the terrestrial radiation emitted by the surface, atmosphere and clouds. In total and in equilibrium, the multi-year global mean sum should be zero. However, as climate change continues and the amount of greenhouse gases in the atmosphere increases, this effect exceeds zero, i.e. more radiation is trapped in the atmosphere than is emitted leading to global warming. In climate modelling, the amounts of radiative

energy absorbed and emitted in and by the atmosphere are key factors in the Earths energy balance. The concentration of water vapour in the atmosphere and the surface albedo are important factors (Loeb et al., 2009). It is important to reproduce these factors correctly and to detect possible biases. The radiation fluxes are calculated by the MESSy submodel RAD, which is a new implementation of the ECHAM5 radiation scheme (Dietmüller et al., 2016). Rad$_{\mathrm{TOA}}$ is compared to the ERA5 monthly averaged reanalysis data of TOA solar and terrestrial radiation interpolated to the EMAC T63 grid (Hersbach, 2023).

The average temporal and spacial correlation between ERA5 and EMAC/JSBACH Rad$_{\mathrm{TOA}}$ is 0.907. The largest differences and overestimation of EMAC/JSBACH Rad$_{\mathrm{TOA}}$ during the summer months occur over North and West Africa and the Middle East (**Fig. 9**). The best agreement is found during summer over western Russia. The largest underestimations are found over Central Africa and Northern South America. During winter, the largest overestimation occurs over the Himalayas and the largest underestimation over the northern and southern Andes, Central Africa and Indonesia. During this time period, the best

agreement is found over the polar and sub-polar regions. The average annual global difference between EMAC/JSBACH and ERA5 is $3.56\,Wm^{-2}$. The zonal mean of the simulation is well in line with the zonal mean of the reanalysis data, and the overestimation of the simulation only occurs at $30°N$ and between $0-30°S$. Comparing EMAC/SRF and ERA5 a similar geographical distribution is discernable and especially in winter there is almost no difference to the EMAC/JSBACH simulation. However, almost everywhere at high latitudes the TOA radiative flux is lower during summer. The same applies for the zonal

average. The average annual global difference between EMAC/SRF and ERA5 is $3.045\,Wm^{-2}$ and the overall correlation 0.909.

The TOA radiation derived from EMAC/JSBACH shows noticeable regional variations when compared to reanalysis data, yet its overall balance remains comparable to the one derived from EMAC/SRF. Moreover, these regional differences in Rad$_{\mathrm{TOA}}$ closely align with those observed for EMAC/SRF and do not significantly change when EMAC is operated with-

out JSBACH. Rad$_{\mathrm{TOA}}$ shows a strong correlation to the surface albedo ($\rho = -0.86$), which determines the amount of absorbed radiation at the surface (**Appendix Table A2**). Since there are no significant differences between the EMAC/SRF and EMAC/JSBACH surface albedo, no significant differences in Rad$_{\mathrm{TOA}}$ are expected as long as atmospheric dynamics, cloud occurrence, and chemical composition of the atmosphere remain the same.





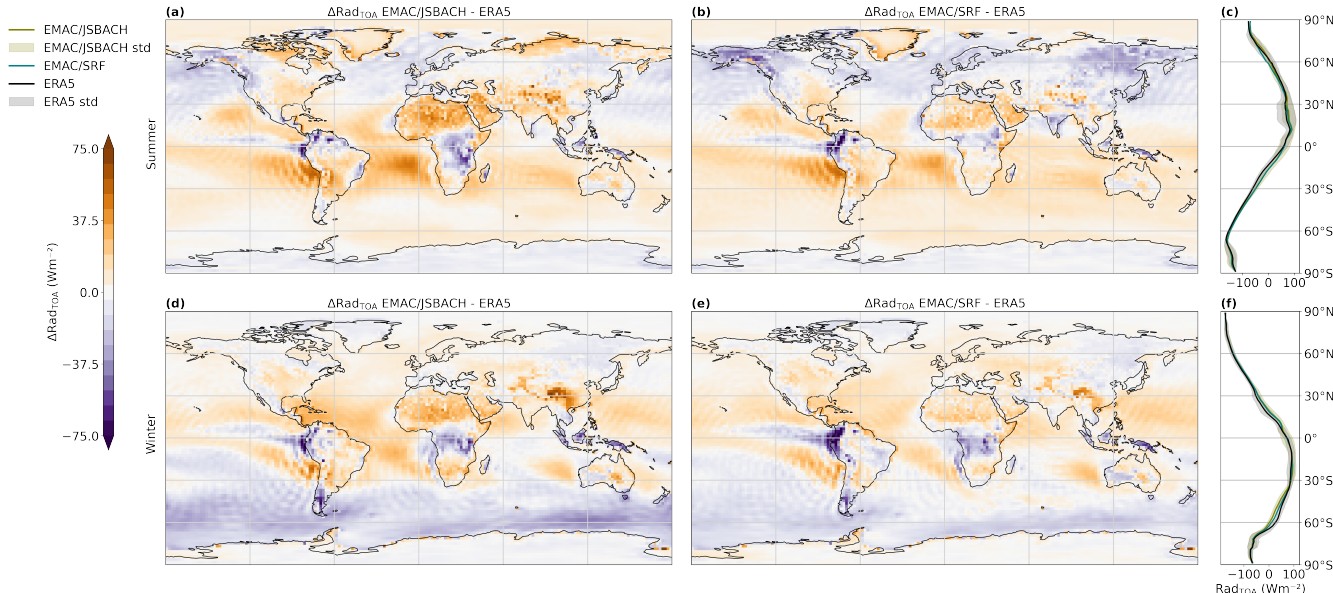

**Figure 9.** Difference of top of atmosphere radiation ($Rad_{TOA}$) between EMAC/JSBACH and ERA5 during Northern Hemispheric (NH) summer (a) and NH winter (d) months, with data averaged over the years 1971 to 2010. Analogously the difference between EMAC/SRF and ERA5 $Rad_{TOA}$ during summer (b) and winter (e) months is displayed. Positive values represent an overestimation of the simulated $Rad_{TOA}$, while negative values indicate an underestimation. Additionally, the zonal average of all three datasets for both summer (c) and winter months (f) is shown. Here, $Rad_{TOA}$ from EMAC/JSBACH is depicted in green, $Rad_{TOA}$ from EMAC/SRF is shown in blue, and $Rad_{TOA}$ from the ERA5 dataset is represented in black. The shaded area within the zonal mean plot illustrates the standard deviations along longitudes.



## 4.5 Precipitation (precip)

Since precipitation is one of the most important and challenging climate variables to reproduce for coupled global climate
models (Dai, 2006), we are interested to analyse the general performance of the coupled EMAC/JSBACH and EMAC/SRF
simulations to reproduce regional and temporal variations as well as the amount and intensity of precipitation. Problems of the
simulation of precipitation can be an indication of issues of the processes that drive precipitation, such as large- and small-scale
atmospheric dynamics, cloud micro-physics, aerosol formation, and many others (Dai, 2006). Precipitation is calculated by the
submodels CLOUD and CONVECT and is one of the standard input parameters for EMAC/JSBACH, forcing many processes
in the land system. The simulated precipitation is compared to the Global Precipitation Climatology Project (GPCP) data-set
of monthly precipitation spanning 1979 to 2021 (Adler et al., 2003). The observational precipitation data is available at a grid
resolutin of $2.5°\,x\,2.5°$, approx. 280 km at the Equator and is regridded to the EMAC Gaussian T63 grid ($1.88°\,x\,1.8°$ approx.
$210\,km$ at the Equator) using bi-linear interpolation. The dataset provides an error estimate, which is defined for every data
point present in the dataset. This assessment considers solely the stochastic error and relies on both, the mean rainfall rate and
the quantity of samples utilized for its computation (Huffman, 1997).

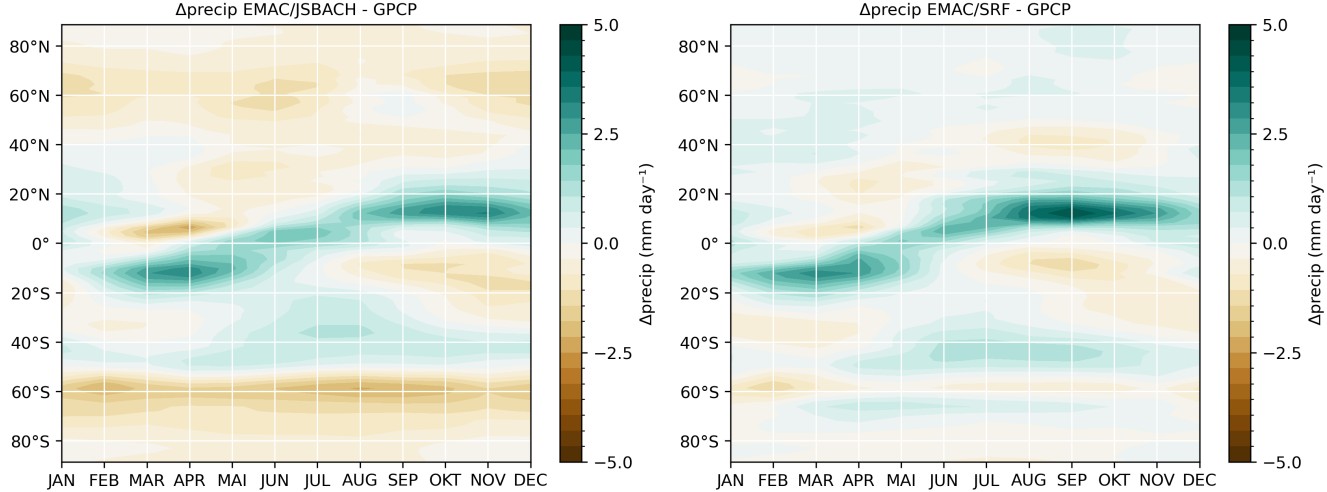

**Figure 10.** Zonally averaged monthly difference of EMAC/JSBACH (a) and EMAC/SRF (b) vs. GPCP precipitation in $mm\,day^{-1}$ averaged over the years 1980 to 2010.

The differences between global precipitation distributions are shown in **Fig. 11**. The subtropical and tropical equatorial zones
over land appear drier in the simulation results in comparison to the GPCP data. This shifts from the Northern Hemisphere
during summer to the Southern Hemisphere during winter. Over the oceans, regions of heavy precipitation are intensified in
the simulation. The average annual global difference between EMAC/JSBACH and ERA5 is $0.042\,mm\,day^{-1}$. For the GPCP
dataset, the error is shown as a grey shade within the zonal mean plot. Zonally averaged, EMAC/JSBACH summer precipitation



is within the GPCP precipitation error for the tropics and mid-latitudes, with EMAC/JSBACH underestimating precipitation at high latitudes. In the tropical region of the Northern Hemisphere, EMAC/JSBACH also slightly overestimates precipitation during winter. The Northern Intertropical Convergence Zone (ITCZ) is reproduced in agreement with the observations by the EMAC/SRF and EMAC/JSBACH simulations. The comparison between EMAC/SRF and GPCP shows that the simulated precipitation amounts are higher than the observed ones, with an average annual global deviation of $0.329\,mm\,day^{-1}$. Similar to EMAC/JSBACH, this simulation shows lower precipitation over Indonesia compared to the GPCP data. Particularly noticeable is the tendency to overestimate precipitation over land, especially in the Himalayas in summer and in the Andes in winter. This is also evident in the zonal mean, although EMAC/SRF remains within the margin of error of the observations in summer. The only exception is the tropical regions of the Northern Hemisphere, where there is more precipitation in winter and less around $60\,^\circ S$.

Overall EMAC/JSBACH is capable of reproducing global precipitation to a similarly extent as EMAC/SRF. It exhibits a distinct wet bias zone, extending from $20\,^\circ S$ to the equator during the winter and from $20\,^\circ N$ to the equator during the summer (**Fig. 10a**). This wet bias band is adjacent to a dry bias region, ranging from $0-10\,^\circ N$ during NH winter and from $0-25\,^\circ S$ during NH summer. The same is found for the precipitation derived from EMAC/SRF (**Fig. 10b**) and leads to the assumption that there is no major change of the large scale atmospheric dynamics of the new coupled model. The wet bias will be corrected by further "tuning" of microphysical parameters in upcoming model versions. The dry biases over the Arctic and Antarctic regions throughout the year may be attributed to relatively low sea surface temperatures. This relationship has previously been documented by Pozzer et al. (2011b).





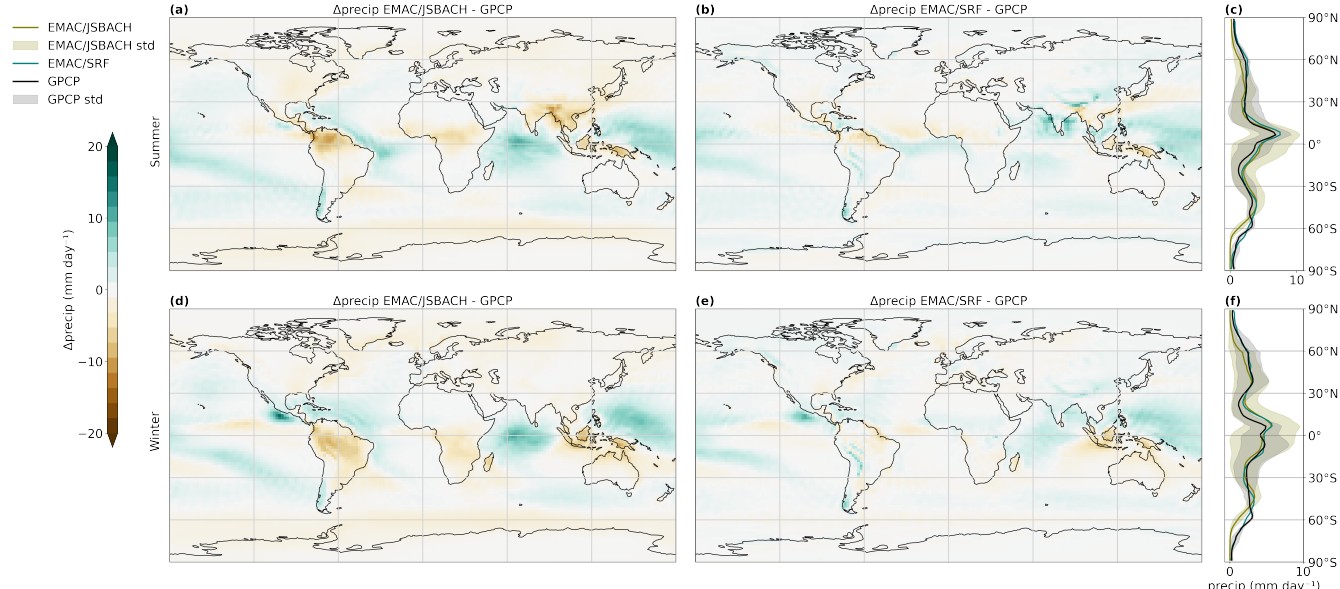

**Figure 11.** Difference of precipitation between EMAC/JSBACH and GPCP during Northern Hemispheric (NH) summer (a) and NH winter (d) months, with data averaged over the years 1980 to 2010. Analogously the difference between EMAC/SRF and GPCP precipitation during summer (b) and winter (e) months is displayed. Positive values represent an overestimation of the simulated precipitation, while negative values indicate an underestimation. Additionally, the zonal average of all three datasets for both summer (c) and winter months (f) is shown. Here, precipitation from EMAC/JSBACH is depicted in green, precipitation from EMAC/SRF is shown in blue, and precipitation from the GPCP dataset is represented in black. The shaded area within the zonal mean plot in black illustrates the GPCP dataset error and in green the standard deviation of the EMAC/JSBACH precipitation.



## 4.6 Leaf Area Index (LAI)

The leaf area index is defined by Watson (1947) as the total one-sided area of leaf tissue per unit ground surface area. Especially for estimating the gas exchange between vegetation and the atmosphere (eg. photosynthetic production or transpiration) it is an important quantity and represents the canopy-atmosphere interface (Bréda, 2003). It has a strong spatial and temporal variability, which makes it difficult to properly measure and to simulate it. The default scheme to calculate the LAI in JSBACHv4 is an implementation of the Logistic Growth Phenology model (LoGro-P model) which is described in detail by Böttcher et al. (2016); Reick et al. (2021). Within LoGro-P the LAI is calculated depending on phenology type of the plant functional types (PFT), which are either evergreen, summergreen, raingreen, grasses, tropical or extratropical crops. Due to the prescribed geographical PFT distribution, there is no seasonal or inter-annual variability in plant functional types, limiting LAI variability. The phenology types are linked to certain phenology phases. For the summergreen type these are in turn associated with seasons. Spring corresponds to the growth phase, summer to the vegetative phase, and winter and autumn to the resting phase (Schneck et al., 2022). Raingreen, grasses, and tropical crop phenology types are only linked to a growth phase determined by environmental conditions like soil moisture, temperature and NPP, growing whenever those conditions are beneficial (Schneck et al., 2022). Tropical or extratropical crops are not linked to a vegetative phase. The LAI changes primarily due to temperature, soil moisture and NPP, and the maximum possible value is limited for each phenology type individually (Schneck et al., 2022). The LAI is compared to MOD15A2H MODIS/Terra Leaf Area Index/FPAR 8-Day L4 Global 500m SIN Grid V061 regridded to global data at 0.5 resolution (Kern, 2021; Running et al., 2015). Monthly averages are calculated and interpolated to the EMAC T63L31 grid. The LAI difference between the EMAC/JSBACH and MODIS observations is shown in **Fig. 12**. The annual global average LAI within EMAC/JSBACH is $-0.212\,m^2\,m^{-2}$ lower than the one estimated from MODIS satellite data. In particular, the LAI of tropical rainforests is underestimated throughout the year in the simulation, as are deciduous forests and boreal forests over eastern Siberia in summer. Otherwise, vegetation LAI tends to be overestimated with peaks in India, South Africa and northern Canada throughout the year and in northern Europe during the winter. The zonal average shows that EMAC/JSBACH follows the MODIS LAI trend, but with lower peak values at the equator. EMAC/SRF overestimates LAI in comparison to the MODIS data-set almost everywhere, and the annual global average LAI is $0.768\,m^2\,m^{-2}$ larger than the satellite instrument based estimate. Maximum overestimations are found for the Amazon rain forest and Canadian boreal forest throughout the year. The same is found for the zonal average, of which the EMAC/SRF LAI peaks at the equator at $7.5\,m^2\,m^{-2}$.

Discrepancies between EMAC/SRF and EMAC/JSBACH LAI are expected, due to EMAC/SRF's reliance on a LAI climatology, whereas in EMAC/JSBACH LAI is a prognostic variable. In JSBACH, the calculation of LAI for raingreen and crop phenology strongly depends on water availability. Tropical raingreen phenology is found in regions such as the Amazon, Indonesia, and Central Africa. These regions exhibit low TWS and coincide with regions of underestimated LAI. Over India the phenology only consists of tropical broadleaf deciduous forests and both, C3 and C4 crops. Given that the water deficit in India is not as pronounced as in other areas, the overestimation of LAI in India may be partly attributed to sufficient moisture content in the soil. In addition, GPP is enhanced in this area. This results in a feedback loop, as more vegetation leads to





greater LAI, which in turn increases GPP and NPP, thus stimulating plant growth. The overestimation of LAI of extratropical
445    evergreen and summergreen phenology, as in northern Canada and Europe, is not determined by water availability, since those
LAI calculations are only based on parameterizations governing phenology and a set of parameters defining growth rate and the
length of the growth season. Schneck et al. (2022) stated that an unlucky choice of those parameters can have a major effect on
LAI estimation. However Lin et al. (2023) found that the MODIS Version 6.1 Leaf Area Index product tends to underestimate
LAI particularly in northern latitudes, which may contribute to the bias found over northern Canada and Europe.

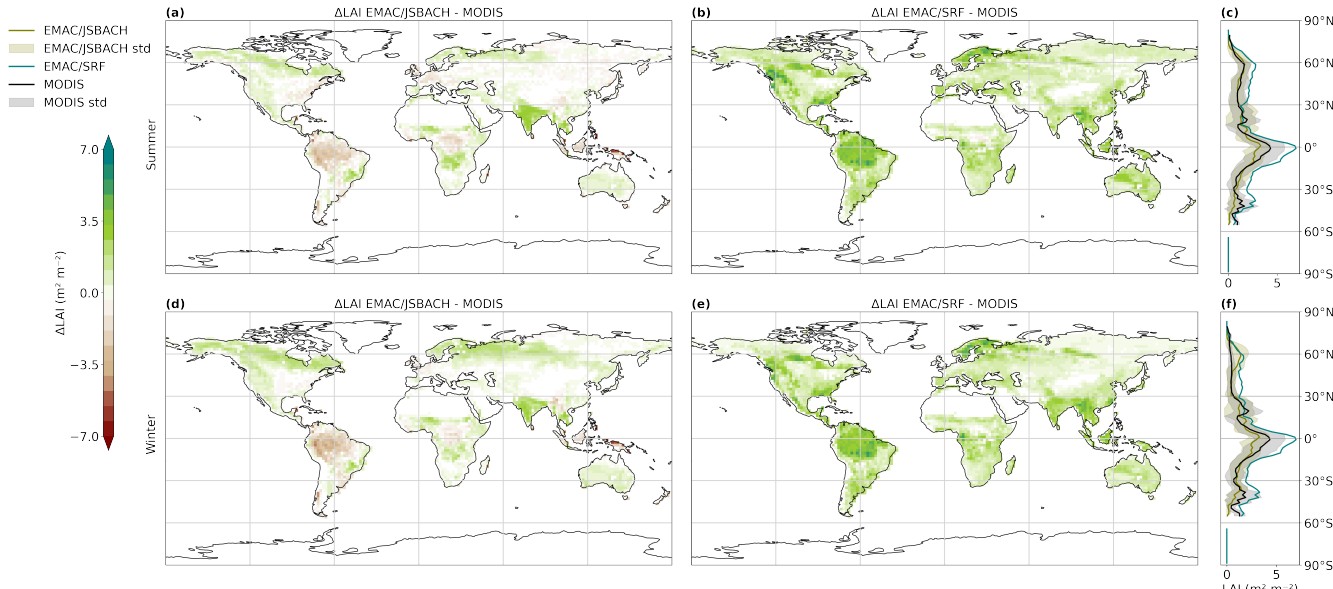

**Figure 12.** Difference of LAI between EMAC/JSBACH and MODIS during Northern Hemispheric (NH) summer (a) and NH winter (d)
months, with data averaged over the years 2000 to 2010. Analogously the difference between EMAC/SRF and MODIS LAI during summer
(b) and winter (e) months is displayed. Positive values represent an overestimation of the simulated LAI, while negative values indicate an
underestimation. Additionally, the zonal average of all three datasets for both summer (c) and winter months (f) is shown. Here, LAI from
EMAC/JSBACH is depicted in green, LAI from EMAC/SRF is shown in blue, and LAI from the MODIS dataset is represented in black. The
shaded area within the zonal mean plot illustrates the standard deviations along longitudes.



## 4.7 Fraction of absorbed photosynthetic active radiation (FaPAR)

Together with the LAI, the fraction of absorbed photosynthetic active radiation is required to estimate the ecosystem productivity. It is a state variable describing the amount of incoming solar radiation absorbed by leaves and available for photosynthesis. The absorption happens in the photosynthetic active radiation (PAR) band of $400 - 700 \, nm$ (Sellers, 1985) and depends on the solar zenith angle, canopy thickness, type of leaves, their optical properties, the orientation, and the soil underneath (Reick et al., 2021).

In EMAC/JSBACH faPAR is calculated in three canopy layers by the canopy radiation module, which is in detail described by Loew et al. (2014) and Reick et al. (2021). After the calculation, faPAR is handed over to the photosynthesis module and used to estimate the gross and net primary productivity and carbon fixation in the plants.

We compare our results to MOD15A2H MODIS/Terra Leaf Area Index/FPAR 8-Day L4 Global 500m SIN Grid V061 regridded to global data at 0.5 resolution derived by ICDC (Kern, 2021; Running et al., 2015). Monthly averages are calculated and interpolated to the EMAC T63 grid.

The fraction of absorbed photosynthetic active radiation (faPAR) is a newly introduced variable that was not available as EMAC output before the coupling to JSBACH. When compared to MODIS, the simulated faPAR in EMAC/JSBACH is systematically underestimated across most regions, with the exception of India and northern Canada (**Fig. 13**). This underestimation is also evident for the zonal average. The annual global average faPAR simulated by EMAC/JSBACH is $0.161 \pm 0.16$, whereas MODIS data indicate a higher average of $0.384 \pm 0.25$. Disney et al. (2016) conducted a comparison between the MODIS product and faPAR and LAI measurements obtained from the ESA GlobAlbedo product. GlobAlbedo aligns with the 1D - radiative transfer schemes used in EMAC/JSBACH and other large-scale ESMs. Their findings indicated overall good agreement in terms of timing between the datasets. Nevertheless, notable discrepancies in peak values were detected, with GlobAlbedo-derived values generally registering lower compared to MODIS. They also state that the method used to determine faPAR can result in variations of up to an order of magnitude difference. Loew et al. (2014) found a difference of up to 25% when comparing models and satellite observations. These biases can be attributed in parts to uncertainties in total cloud cover and snow cover that may affect satellite based measurements. However, they may also result from the underlying definitions and algorithms used to determine the faPAR product from satellite instruments (Loew et al., 2014). Within EMAC/JSBACH, the largest potential cause of uncertainty is the bias of the LAI, which for example most likely leads to the overestimation of faPAR over India. Additionally, it cannot be ruled out that the representation of surface albedo and the radiative transfer scheme might lead to the general underestimation of faPAR, as was also documented in former studies (Loew et al., 2014; Disney et al., 2016). The differences of EMAC and observations in the radiative fluxes (mentioned above), especially the shortwave components, might also substantially contribute to the bias in faPAR.



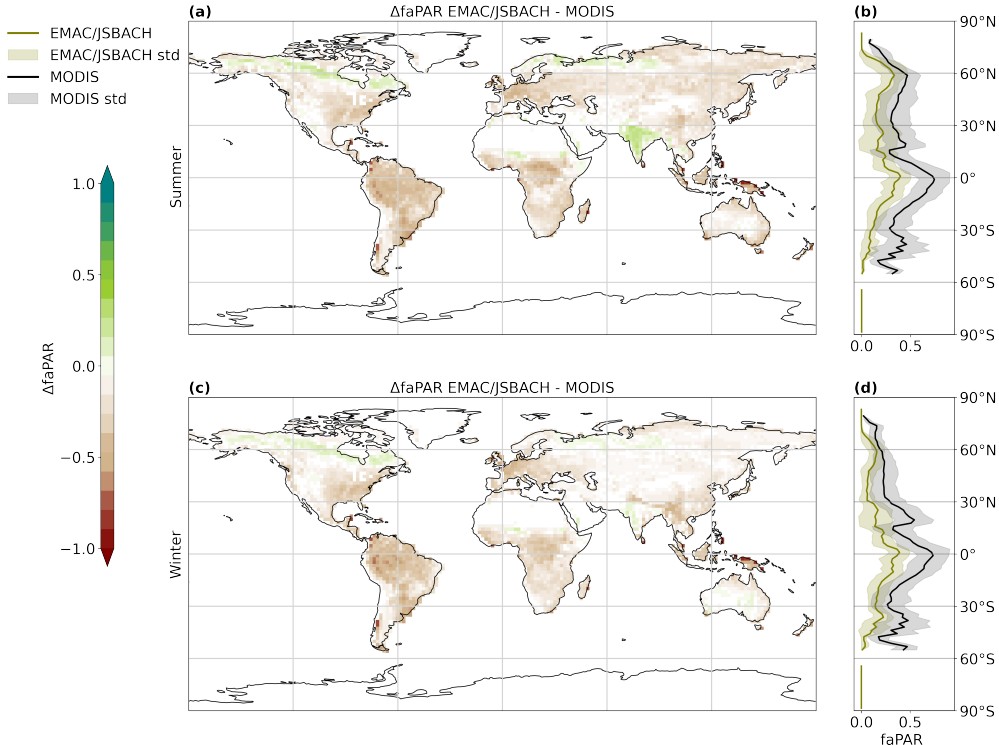

**Figure 13.** Difference of fraction of absorbed photosynthetic active radiation (FaPAR) between EMAC/JSBACH and MODIS during Northern Hemispheric (NH) summer (a) and NH winter (c) months, with data averaged over the years 2000 to 2010. Positive values represent an overestimation of the simulated FaPAR, while negative values indicate an underestimation. Additionally, the zonal average of both datasets for both, summer (b) and winter months (d), is shown. Here, FaPAR from EMAC/JSBACH is depicted in green and FaPAR from the MODIS dataset is represented in black. The shaded area within the zonal mean plot illustrates the standard deviations along longitudes.

## 4.8 Gross Primary Productivity (GPP)

Gross Primary Productivity is the total rate of organic carbon gained via photosynthesis. This includes autotrophic respiration, which can be divided in maintenance respiration (driving basic functionalities of the plant like water and nutrient transport, defence mechanisms or repairs), and the growth respiration. GPP is a key parameter to estimate the Net Primary Productivity (NPP), which describes the actual amount of carbon (sugars) stored in vegetation and is, therefore, an important quantity for the terrestrial carbon cycle. It is highly dependent on radiation, temperature, precipitation, LAI, TWS and water usage efficiency (the amount of water used by the plant to assimilate carbon). In EMAC/JSBACH GPP is calculated via carbon assimilation (based on the plant water stress), faPAR and LAI. The full and detailed description of the dynamics of vegetation carbon is provided by Reick et al. (2021).



GPP is compared to the MOD17A2H MODIS/Terra Gross Primary Productivity 8-Day L4 Global 500m SIN Grid V006
regridded to global data at 0.5 resolution derived by ICDC (Kern, 2021; Running et al., 2015). Monthly averages are calculated
and interpolated to the EMAC T63 grid.

Similar to faPAR, Gross Primary Productivity is a new diagnostic introduced in EMAC by JSBACH. GPP shows the largest
differences with MODIS observations during NH summer over India, where GPP is strongly overestimated (**Fig. 14**). This
is slightly less during winter, when the GPP overestimation is larger over Australia and central South America. The largest
underestimation is found during summer month over northeastern Siberia, the Amazon region and central Africa, while dur-
ing summer the largest underestimation is found in the Amazon basin and the Andes. The annual global average GPP of
EMAC/JSBACH is $0.02 \pm 0.02 \, kg \, Carbon \, km^{-2}$, while that from MODIS is $0.021 \pm 0.02 \, kg \, Carbon \, km^{-2}$.

GPP underestimations are mostly found in areas where TWS and faPAR are also underestimated. The correlation between
TWS and GPP is $\rho = 0.83$, indicating a monotonic relationship (**Appendix Table A2**). The correlation between faPAR and
GPP is, as expected, high with $0.94$, since the GPP calculation is based on faPAR. However, in this study, GPP demonstrates
a notably better agreement with MODIS observations than faPAR. The pronounced overestimation of GPP over India can be
largely attributed to the concurrent overestimation of faPAR in that region, which, in turn, can be traced back to the high LAI
values prevalent there. In the global mean, faPAR derived from EMAC/JSBACH and MODIS are in good agreement.



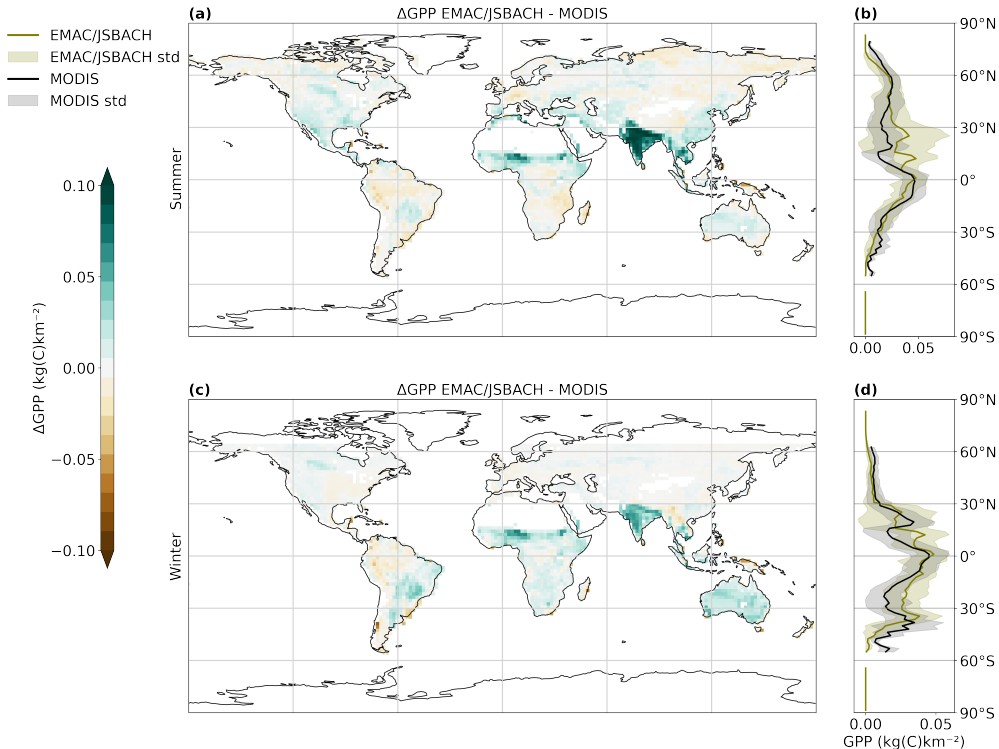

**Figure 14.** Difference of gross primary productivity (GPP) between EMAC/JSBACH and MODIS during Northern Hemispheric (NH) summer (a) and NH winter (c) months, with data averaged over the years 2000 to 2010. Positive values represent an overestimation of the simulated GPP, while negative values indicate an underestimation. Additionally, the zonal average of both datasets for both, summer (b) and winter months (d), is shown. Here, GPP from EMAC/JSBACH is depicted in green and GPP from the MODIS dataset is represented in black. The shaded area within the zonal mean plot illustrates the standard deviations along longitudes.



## 5   Conclusions

We have implemented the land surface model JSBACH version 4 as a new submodel into EMAC, following the MESSy coding standards. The new addition lows for a replacement of the former, simplified submodel SURFACE in which many parameters have been prescribed based on pre-determined climatologies. JSBACH comprises numerous new features, including a comprehensive hydrology model and an improved soil scheme enhancing the overall versatility of EMAC. It enables the possibility to perform new experiments analysing not only fundamental physical processes of the land surface within the Earth
system on climatic time scales, but also the effects of atmospheric chemical components and associated feedback mechanisms on short time scales of hours and days. In this assessment, we demonstrate that the implementation, various modifications, and newly added features to EMAC have not degraded the overall model performance and stability. This is done based on a comparison of the new coupled model results to observational and reanalysis data, and in comparison to results from a simulation conducted without JSBACH (i.e., based on climatologies). The newly coupled land-atmosphere model, however,
required re-tuning to optimise the radiation budget via adjusted cloud parameters. The usage of JSBACH instead of SURFACE increases the runtime on average by $0.056\% \pm .4e-05\%$ for simulations carried out on two compute nodes of the DKRZ (Deutsches Klimarechenzentrum) supercomputer LEVANTE.

Results indicate that the LST derived from the newly coupled model is on average $1.546°K$ colder compared to the LST derived from ERA5. The change from SURFACE to JSBACH improves the representation of TWS by generally increasing
soil moisture and groundwater storage. This improves the agreement of the absolute global average TWS with the reanalysis data and reduces the NRMSE. Surface albedo and $\text{Rad}_{\text{TOA}}$ balance show no significant changes after the implementation of JSBACH. While seasonal and regional precipitation patterns are preserved, the global mean precipitation is slightly reduced in EMAC/JSBACH. The average global LAI of the EMAC/JSBACH simulation agrees better with the average LAI of MODIS than the climatological standard LAI present in EMAC/SRF; nevertheless, the spacial and temporal correlation of $0.637$ be-
tween simulated LAI and observed LAI is still not very high. FaPAR and GPP are among many other newly introduced variables that were not available in previous EMAC versions. They are now provided as diagnostic parameters. FaPAR shows the largest deviation from the observations, which could be partly due to challenges in observing and quantifying faPAR. Nevertheless, faPAR as a fundamental parameter within the GPP calculations seems realistic, as the GPP and observational global average difference is only $-0.001\,kg\,Carbon\,km^{-1}$.

We plan to implement the remaining JSBACH4 features, such as the closes carbon cycle and dynamic vegetation. The latter can be achieved until these updates are available by linking JSBACH with the dynamic vegetation of the LPJ-GUESS module already coupled in EMAC (Forrest et al., 2020). The model will be further refined to increase its capabilities and accuracy. This ongoing model development is crucial towards the more comprehensive and realistic numerical modelling of the intricate interactions between the atmosphere and land, along with the associated feedback mechanisms. It marks a significant
advancement for EMAC, bringing it one step closer to the realisation of a comprehensive Earth System Model.



*Code availability.* The Modular Earth Submodel System (MESSy, https://doi.org/10.5281/zenodo.8360186) is continuously further developed and applied by a consortium of institutions. The usage of MESSy and access to the source code is licenced to all affiliates of institutions which are members of the MESSy Consortium. Institutions can become a member of the MESSy Consortium by signing the MESSy Memorandum of Understanding. More information can be found on the MESSy Consortium Website (http://www.messy-interface.org). The code

presented/used here is available from https://doi.org/10.5281/zenodo.10084186 and will be part of the next official release. It has been based on MESSy version d2.55.2 and JSBACH version 4 available via the jsbach Repository on GitLab (#7de0f9bf3b50910655f474bc23d647c6ba2a7b6f).





# Appendix A

## A1

**Table A1.** EMAC/JSBACH Land cover types (lct) and corresponding tile in the EMAC/JSBACH simulation (grey). LCT's without tile indication are not used in the 11 tile setup.

| LCT | Description | Tile |
|-----|-------------|------|
| lct01 | "glacier" | 1 |
| lct02 | "tropical broadleaf evergreen" | 1 |
| lct03 | "tropical broadleaf deciduous" | 2 |
| lct04 | "extra-tropical evergreen" | 3 |
| lct05 | "extra-tropical deciduous" | 4 |
| lct06 | "temperate broadleaf evergreen trees" | |
| lct07 | "temperate broadleaf deciduous trees" | |
| lct08 | "coniferous evergreen trees" | |
| lct09 | "coniferous deciduous trees" | |
| lct10 | "raingreen shrubs" | 5 |
| lct11 | "deciduous shrubs" | 6 |
| lct12 | "C3 grass" | 7 |
| lct13 | "C4 grass" | 8 |
| lct14 | "pasture" | |
| lct15 | "C3 pasture" | 9 |
| lct16 | "C4 pasture" | 10 |
| lct17 | "tundra" | |
| lct18 | "swamp" | |
| lct19 | "crops" | |
| lct20 | "C3 crops" | 11 |
| lct21 | "C4 crops" | 11 |

## A2





**Table A2.** Spearman rank correlation ($\rho$) of the assessed variables derived from monthly means of EMAC/JSBACH for the years 1971 to 2010. A positive Spearman rank correlation suggests a monotoneously increasing relationship, while a negative correlation indicates a monotoneously decreasing relationship. All correlations were tested for statistical significance at the $p < 0.05$ level.

|  | LST | TWS | Surface Albedo | $Rad_{TOA}$ | Precipitation | LAI | faPAR | GPP |
|---|---|---|---|---|---|---|---|---|
| LST | 1.0 |  |  |  |  |  |  |  |
| TWS | 0.75 | 1.0 |  |  |  |  |  |  |
| Surface Albedo | -0.89 | -0.80 | 1.0 |  |  |  |  |  |
| $Rad_{TOA}$ | 0.87 | 0.66 | -0.86 | 1.0 |  |  |  |  |
| Precipitation | 0.75 | 0.80 | -0.87 | 0.68 | 1.0 |  |  |  |
| LAI | 0.67 | 0.77 | -0.79 | 0.66 | 0.78 | 1.0 |  |  |
| faPAR | 0.75 | 0.83 | -0.85 | 0.74 | 0.84 | 0.95 | 1.0 |  |
| GPP | 0.84 | 0.83 | -0.90 | 0.84 | 0.82 | 0.91 | 0.94 | 1.0 |

*Author contributions.* AM and AP planned the research. AM developed the model code and performed the simulations with help of AP. AP, VG and PJ contributed to the overall model development. KK provided the MODIS datasets. BS provided the ERA5 datasets. AM wrote the manuscript with help of AP, VG, PJ, KK, BS, HT and JL. AP, HT and JL supervised the project. All authors discussed the results and contributed to the review and editing of the manuscript.

*Competing interests.* At least one of the (co-)authors is a member of the editorial board of Geoscientific Model Development. The authors
have no other competing interests to declare.

*Acknowledgements.* The model simulations have been performed at the German Climate Computing Centre (DKRZ) through support from the Max Planck Society.



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
