# Peer review of "Evaluation of the coupling of EMACv2.55 and the land surface and vegetation model JSBACHv4"

_EGUsphere, 2023_

## Referee Comment (RC1)

Review of **"Evaluation of the coupling of EMACv2.55 and the land surface and vegetation model JSBACHv4"** by Martin et al.

**General comments:**
This paper examines the performance of the ECHAM/MESSy Atmospheric Chemistry (EMAC) model (v2.55) after implementing the Jena Scheme for Biosphere-Atmosphere Coupling in Hamburg (JSBACH) land surface model (v4). Several key variables affecting water, carbon, and energy fluxes at the land-atmosphere interface are assessed in comparison with observational and reanalysis datasets. The performance of the new EMAC/JSBACH model is also compared with the default version of the model (EMAC/SURFACE). It is found that the newly added features did not degrade the overall performance of the model while greatly improving the representation of land hydrology. Overall, the paper is quite well-written, and I enjoyed reading it. I have a few suggestions for the authors to consider.

**Specific comments:**
1. Section 3.1.1 introduces the reference datasets (e.g., Table 5). It would be informative to also include a brief discussion about the uncertainties associated with these datasets/variables, if possible.

2. While terrestrial water storage (TWS) reflects the performance of land hydrology, I wonder if it's also helpful to examine surface soil moisture and evapotranspiration in the model.

3. For land surface temperature (LST), it would be interesting to include a discussion about why the latent heat fluxes in EMAC/JSBACH are somewhat overestimated (lines 264-269). For instance, does the overestimated TWS partially contribute to this? Also, latent heat alone may not be sufficient to explain LST. I wonder if other energy fluxes, such as surface shortwave and longwave radiative fluxes, and sensible heat are examined as well.

4. Section 2.1, consider adding information about soil layers and their depths.

5. Line 97-108, I wonder if it's possible to include a schematic to demonstrate these processes.

6. Line 161, no values in Table 3 are shown in bold...

7. Line 175, are aerosol concentrations prescribed?

8. Line 291, the soil depth in the ERA5 is much shallower than the EMAC/JSBACH, 2.89 m vs 9.8 m. How does this affect the comparison of TWS?

9. Line 372, "cloud occurrence... remain the same", the differences in LST and latent heat may affect cloud distribution.

10.  In terms of TOA fluxes, have you considered using CERES? Or are ERA5 TOA fluxes assimilated with observations?

11. Line 408, are the prescribed SSTs the same in the EMAC/JSBACH and EMAC/SURFACE runs?

**Technical corrections:**
1. Line 240, 0.1° by 0.1°?

2. Fig. 3 captions, "LST trend" is somewhat misleading as no trends are calculated. Maybe "LST time series"?

---

## Referee Comment (RC2)

Anna Martin and her co-authors implemented the land surface and vegetation model JSBACHv4 into the ECHAM/MESSy Atmospheric Chemistry (EMAC) model. JSBACH is now available as an independent MESSy submodule and can be used instead of the previous soil water bucket model, which is integrated in the SURFACE submodel.

The authors describe how they integrated JSBACH into MESSy and also describe the various processes of JSBACH and how these are divided into different subroutines/functions/files in the MESSy infrastructure.

For the optimization of the previously used parameters in SURFACE, tuning was carried out with the help of numerous EMAC simulations.

A detailed evaluation of JSBACH in comparison to SURFACE was carried out using different observation and reanalysis data (MODIS, GPCP and ERA5-Land). The evaluation shows that JSBACH achieves significantly better results in some cases. In particular TWS, LAI and the precipitation are now closer to the observational data (at least the mean). Other physical variables such as LST or albedo are simulated similarly as with SURFACE. In addition, JSBACH now includes new physical variables such as FaPAR, GPP or NPP, which allow additional accuracy of the soil properties and in particular the properties of the vegetation.

All in all, the authors show a very successful integration of JSBACH into the MESSy infrastructure. EMAC and probably also IMAC (ICON/MESSy), are thus taking a further successful step towards a holistic comprehensive earth system model.

The paper is therefore of great scientific importance and significance.

The scientific quality of the paper is very good. It is written clearly and has a logical and structured structure. The model simulations and evaluations are well described and reasonable. The description of JSBACH in EMAC is very well described and the necessary evaluation is presented in detail. All EMAC simulations and evaluations should be easy reproducible. The complete model code is available in a GitLAB at DKRZ.

I can therefore recommend the paper for publication in any case and have only a few comments that should be taken into account (although many of them are only suggestions, where the authors can decide for themselves whether they want to implement them):

**Comments/Remarks/Suggestions:**

Line 2: "the soil water bucket model" → "the soil water bucket model included in the SURFACE submodel"

Line 8: "coupled model" -> "new coupled model (EMAC/JSBACH)"

Line 9: "MODIS" → "Moderate Resolution Imaging Spectroradiometer (MODIS)"

Line 15: I would move the sentence "The LAI climatology in EMAC has been substituted with a refined method for directly calculating LAI" to line 6

Lines 16/17: I would remove here the sentence "FaPAR and GPP exemplify two of the many additional variables made available through JSBACH in EMAC", but insert a sentence in the area of line 4 to 7, describing which variables are now new in EMAC (not included in SURFACE before).

Lines 20 to 23: "This improvement can be attributed to a general increase in soil moisture and water storage in deeper soil layers, leading to a reduction in normalised root mean square error (NRMSE) and a closer alignment of simulated TWS with observations, mitigating the previously widespread problem of soil drought." → "This improvement can be attributed to a general increase in soil moisture and water storage in deeper soil layers, and a closer alignment of simulated TWS with observations, mitigating the previously widespread problem of soil drought."

Line 23: I would either remove the sentence "The correlation of TWS and observations is 0.251 and the average global difference is 0.052m water", because that sounds contradictory (due to the very low correlation and the fact that the means are not mentioned) to the previous sentence or alternatively mention also the means 1.13m (model) and 1.078m (ERA5).

Line 24: "We show that the numerous newly added components strongly improve the land hydrology, e.g. soil moisture; while surface parameters, which were mostly prescribed according to climatologies, remain similar." → "We show that the numerous newly added components strongly improve the land surface, e.g. soil moisture, TWS, and LAI, while surface parameters, as LST, surface albedo or Rad$_{TOA}$, which were mostly prescribed according to climatologies, remain similar."

Line 34: "Gutiérrez et al., 2021". I think this is not the correct citation for the Annex II of IPCC, 2021. The correct citation (also in the literature) is in my opinion an mention in the document itself: IPCC, 2021: Annex II: Models [Gutiérrez, J M., A.-M. Tréguier (eds.)]. In Climate Change 2021: The Physical Science Basis. Contribution of Working Group I to the Sixth Assessment Report of the Intergovernmental Panel on Climate Change [Masson-Delmotte, V., P. Zhai, A. Pirani, S.L. Connors, C. Péan, S. Berger, N. Caud, Y. Chen, L. Goldfarb, M.I. Gomis, M. Huang, K. Leitzell, E. Lonnoy, J.B.R. Matthews, T.K. Maycock, T. Waterfield, O. Yelekçi, R. Yu, and B. Zhou (eds.)]. Cambridge University Press, Cambridge, United Kingdom and New York, NY, USA, pp. 2087–2138, doi:10.1017/9781009157896.016

Line 37: "… (Roeckner et al., 2006)" → "… ,Roeckner et al., 2006)"

Line 37: "However, all physical parameterisations from ECHAM have been replaced …". I find this sentence misleading, as the parameterizations were not replaced in most cases, but rather outsourced to certain submodels and where appropriate supplemented there with further parameterizations. It is better described in the lines 124/125.

Line 46: "(LPJ-GUESS)" –> "(LPJ-GUESS, Forrest et al., 2020)"

Line 49: "documented" → "described"

Line 51: "(Reick et al., 2021)" → "(Reick et al., 2013, 2021)"

Line 59: "SURFACE" →"the MESSy submodel SURFACE"

Line 67: "document" → "describe"

Line 69: "evaluation" → "corresponding evaluation"

Line 73: "Reick et al. (2013)" → "Reick et al. (2013, 2021)"

Line 75: "Briefly summarized, on the technical side …" → "On the technical side …"

Line 86: "The ICON-Land infrastructure allows a clear separation of the physical processes used in JSBACHv4." → "In the case of the ICON-Land infrastructure a clear separation of the physical processes used in JSBACHv4 is allowed."

Line 87: "defined via gross and net primary productivity and photosynthesis" → "defined via gross (GPP) and net primary productivity (NPP) and photosynthesis"

Lines 91/92: Please add a sentence that different land cover types are listed in Table A2.

Line 99: "jsbach" → "JSBACH"

Lines 108/109: I would make a paragraph here (before JSBACH).

Line 129: "In future simulations where JSBACH is used, the SURFACE submodel must be switched off in the namelist setup." → "In the case JSBACH is used, the SURFACE submodel must be switched off."

Line 141: "The simulation based on the default parameters …" → "The simulation using JSBACH based on the default parameters …"

Lines 144/145: "… Table S2 of the Supplement. Simulation 2 and 31 were not completed due to server failures and were excluded from the analysis." →"Table S2 of the Supplement (simulations 2 and 31 were not completed due to server failures and were excluded from the analysis)."

Line 148: "ERA5/ERA5-Land monthly averaged data" → "ERA5-Land monthly averaged data"

Lines 160/161: I my opinion you should write two or three sentences more here regarding the results in Table 3. What criteria were applied to decide which parameter sizes were selected?

Line 173: "standard 11 tile setup" → "standard tile setup"

Lines 207/208: I would make a paragraph here (before "LST")

Line 208: "(shown in red in Fig. 1)". In my case this is not red, rather brown. Perhaps don´t mention the colours, in my opinion this is not necessary. I would write "(see Fig.1 and Table 6)".

Line 209: "0.743 K warmer than REF".  According to Table 6 it has to be "0.816 K".

Line 211: "0.946" → "0.947" and "0.943" →"0.944" (according to Table 6)

Line 212: "-0.012" –> "-0.013". Probably the reason is the rounding …

Line 213: " Wm-2 " → " Wm$^{-2}$ "

Lines 239/240: (Muñoz Sabater, 2019, 2021) → "(Muñoz Sabater, 2019)". The both datasets Muñoz Sabater, 2019 and Muñoz Sabater, 2021 were combined into one data set.

Line 246: "is lower everywhere" → "is lower than REF everywhere"

Line 258: "warmer land surface" → "warmer global land surface"

Line 261: "tropics, subtropics" → "tropics/subtropics"

Line 262: "in both simulations" → "in the EMAC/JSBACH and the EMAC/SRF simulation"

Line 283: Also evaporation?

Line 293: "In Fig. 7 the difference of TWS between EMAC/JSBACH and ERA5 is shown." → "In Fig. 7 the difference of TWS between EMAC/JSBACH and EMAC/SRF to ERA5 is shown."

Line 293ff, Fig.6/Fig.7: I would change the numbers of the two figures, Figure 7 to Figure 6 and Figure 6 to Figure 7, so that they appear in the text in the correct order.

Line 295: "Russia" → "Western Russia"

Line 295: "EMAC/JSBACH overestimates TWS almost everywhere, except for high elevated regions" →"EMAC/JSBACH overestimates TWS almost everywhere, independent of the season, except for high elevated regions"

Line 298: → Fig.7 (right panels)

Line 299: "… than the EMAC/SRF results, which is also visible in Fig. 6." → "… than the EMAC/SRF results. This is also visible in Fig. 6. where the globally averaged TWS trend is illustrated."

Line 300: "The TWS of the EMAC/SRF simulation is lower everywhere …" → "The TWS of the EMAC/SRF simulation (Fig. 7) is lower everywhere …"

Line 302: "−0.68 m lower than the one derived from reanalysis data." →"−0.684 m lower than the one derived from reanalysis data (see Table 6). In EMAC/JSBACH the global average of TWS is 1.13 ± 0.706 m which is, with a difference of 0.052m, significantly closer to ERA5 (1.078±0.56m)."

Line 304: "computed" → "simulated"

Line 310: "0 ◦C" → "0 °C"

Line 325: I would make a paragraph here (after "… (2021).")

Line 327: "?Schaaf and Wang," → "Schaaf and Wang,"

Lines 330-332: "During summer, EMAC/JSBACH shows a slight overestimation in the Northern Hemisphere and an underestimation in the Southern Hemisphere. In the winter months the opposite applies, underestimation in the Northern Hemisphere and overestimation in the Southern Hemisphere." This is only true for different parts of the Northern Hemisphere and Southern Hemisphere. Please be more specific.

Line 336: "Fig.8"→ "Fig.8 (right panels)"

Line 355: "0.907" → "0.907 (Table 6)"

Line 355: "ERA5 and EMAC/JSBACH" →"EMAC/JSBACH and ERA5

Line 356: "… of EMAC/JSBACH $Rad_{TOA}$ during … " → ""… of EMAC/JSBACH RadTOA in comparison to ERA5 during …"

Line 370: "correlation" → "anti-correlation"

Line 387ff/Fig.10/Fig.11: I would also here change the numbers of the two figures, Fig. 10 to Fig. 11 and Fig. 11 to Fig.10, so that they appear in the text in the correct order.

Lines 388: "… in the simulation results in comparison …" → " … in the EMAC/JSBACH simulation in comparison …"

Line 390: "… is 0.042 mm $day^{-1}$. " → "… is 0.042 mm $day^{-1}$ (2.738 to 2.696 mm $day^{-1}$, see Table 6)." This makes it easier to classify the size of the difference.

Lines 427/428: : I would make a paragraph here (after "… grid.")

Line 427: "… is −0.212 $m^2$ $m^{-2}$ lower …" → "is −0.212 $m^2$ $m^{-2}$ (1.187 to 1.399, see Table 6) lower …"

Line 431: "The zonal average shows" –> "The zonal averages (Fig. 12 right panels) shows"

Line 516: "0.056% ± .4e − 05%" → "0.056% ± 4e−04%

Line 518: "Results indicate that the LST derived from the newly coupled model is on average 1.546°K colder compared to the LST derived from ERA5." → "Results indicate that the LST derived from the newly coupled EMAC/JSBACH model is on global average 1.546 K colder compared to the LST derived from ERA5 (using the old SURFACE submodel, the globally averaged LST was 0.816 K warmer)."

Line 520: "the reanalysis" → "the ERA5 reanalysis"

Line 525: "are among many other newly introduced variables". I would list them all here.

Table 1: "NPP" →"Net primary productivity (NPP)"

Table 3: In the case that the abbreviation of a physical quantity (e.g. $HFLX_{sensible}$) are not explained (long form) in the main text, explain them here at least in the caption

Table 6:

- For the precipitation there is the value 2.738 ± 3.382 (EMAC/JSBACH) and 3.025 ± 3.279 (EMAC/SRF). That cannot be, or? The standard deviation is larger than the mean. This would lead to negative precipitation values …
- Maybe this is not so important, but I would prefer it to see the observational data repeated in the 2nd part of the table.

Table A1: Why do you have two times the number 1 and two times the number 11 here?

Table S2 (Supplement):

- It is not totally clear that the numbers in the EMAC/SRF and in the CTRL rows are the default values. Maybe you can write this in the caption. Furthermore, if the value appears in the table, it would also be good to write "default" there. For example in run 4 instead of "0.85" for zasic write "default". Perhaps the latter is also sufficient to make it clear.
- Please also write in the caption that the simulations 2 to 35 were performed with EMAC/JSBACH.
- What is the difference between CTRL and the runs 4, 8, 20, and 22? Maybe I don´t see it, but are these not all the simulation setups? In Table S3 and S4 are the same results in the case of the simulations 4, 8, 20 and 22. But the results differ to the CTRL simulation. Why is this so?
- Why do you write "EMAC/JSBACH" instead of "16" in the case of run 16. That is confusing. If you want to express that this is the best choice of parameters, please also write this in the caption.

Fig. 5:

- "low cloud cover (lcc), medium cloud clover (mcc) and high cloud cover (hcc)" are not shown.
- "The blue background colour indicates values averaged over the polar climate zone (latitudes > 66.5 °), the green background colour indicates values averaged over the temperate climate zone (latitudes between 40 ° and 66.5 °), and the red background colour indicates values averaged over the tropical and subtropical climate zone (latitudes < 40 °)" ⮕ "The upper panels indicates values averaged over the polar climate zone (latitudes > 66.5 °), the mid panels values averaged over the temperate climate zone (latitudes between 40 ° and 66.5 °), and the bottom panels values averaged over the tropical and subtropical climate zone (latitudes < 40 °)."

References:

1) "Gutiérrez, J M., A.-M. T. e., Masson-Delmotte, V., P. Z. A. P. S. C. C. P. S. B. N. C. Y. C. L. G. M. G. M. H. K. L. E. L. J. M. T. M.-T. W.O. Y. R. Y., and (eds.), B. Z.: Annex II: Models, p. 2087–2138, Cambridge University Press, Cambridge, United Kingdom and New York, NY, USA„ https://doi.org/10.1017/9781009157896.016, 2021"

→

„IPCC, 2021: Annex II: Models [Gutiérrez, J M., A.-M. Tréguier (eds.)]. In Climate Change 2021: The Physical Science Basis. Contribution of Working Group I to the Sixth Assessment Report of the Intergovernmental Panel on Climate Change [Masson-Delmotte, V., P. Zhai, A. Pirani, S.L. Connors, C. Péan, S. Berger, N. Caud, Y. Chen, L. Goldfarb, M.I. Gomis, M. Huang, K. Leitzell, E. Lonnoy, J.B.R. Matthews, T.K. Maycock, T. Waterfield, O. Yelekçi, R. Yu, and B. Zhou (eds.)]. Cambridge University Press, Cambridge, United Kingdom and New York, NY, USA, pp. 2087–2138, doi:10.1017/9781009157896.016"

2) "Hersbach, H., B. B. B. P. B. G. H. A. M. S. J. N. J. P. C. R. R. R. I. S. D. S. A. S. C. D. D. T. J.-N.: ERA5 monthly averaged data on singlelevels from 1940 to present, Copernicus Climate Change Service (C3S) Climate Data Store (CDS), https://doi.org/10.24381/cds.f17050d7,615,  2023."

→

"Hersbach, H., Bell, B., Berrisford, P., Biavati, G., Horányi, A., Muñoz Sabater, J., Nicolas, J., Peubey, C., Radu, R., Rozum, I., Schepers, D., Simmons, A., Soci, C., Dee, D., Thépaut, J-N. (2023): ERA5 monthly averaged data on single levels from 1940 to present. Copernicus Climate Change Service (C3S) Climate Data Store (CDS), DOI: 10.24381/cds.f17050d7 (Accessed on DD-MMM-YYYY)"

3) "Muñoz Sabater, J.: ERA5-Land monthly averaged data from 1981 to present, https://doi.org/10.24381/cds.68d2bb3, 2019.

Muñoz Sabater, J.: ERA5-Land monthly averaged data from 1950 to 1980, https://doi.org/10.24381/cds.68d2bb3, 2021"

→

"Muñoz Sabater, J. (2019): ERA5-Land monthly averaged data from 1950 to present. Copernicus Climate Change Service (C3S) Climate Data Store (CDS). DOI: 10.24381/cds.68d2bb30 (Accessed on DD-MMM-YYYY)"

---

## Author Response (AR1)

**Response to Referee#1**

Thank you very much for the very useful and constructive comments and suggestions. Below is a list of all comments with corresponding replies.

**Specific comments:**

1. Section 3.1.1 introduces the reference datasets (e.g., Table 5). It would be informative to also include a brief discussion about the uncertainties associated with these datasets/variables, if possible.

> In case of the ERA5 and ERA5-Land dataset no uncertainty characterization was performed and is therefore not available. The GPCP dataset includes information of the error and standard deviation, which is now included in the analysis. Regarding the MODIS uncertainty the following was added to line 200 of the revised manuscript:
>
> "A comprehensive overview of the limitations and uncertainties of the MODIS data is provided by Disney et al. 2016. The MODIS standard deviation of LAI and FaPAR are displayed in the Appendix (Figure A2 and A2), together with the GPCP precipitation error (Figure A1) "

2. While terrestrial water storage (TWS) reflects the performance of land hydrology, I wonder if it's also helpful to examine surface soil moisture and evapotranspiration in the model.

> Soil moisture is indeed an important variable to consider. However, since the focus is on the performance of the hydrology model rather than on the comparison of soil moisture in the uppermost layers (up to 2.89 m as in the ERA5 data set) only, but on the water content of the entire soil including the root zone and runoff, TWS was chosen as the evaluation variable. Furthermore, a direct comparison between EMAC/SRF and EMAC/JSBACH soil moisture is not possible, as this variable is not available from the bucket model of EMAC/SRF.

> Evapotranspiration is analyzed indirectly in Section 4.1, where the land surface temperature (LST) is analyzed. Since the vegetation cover does not change significantly between the simulations and the leaf area index (LAI) in the EMAC/JSBACH simulation is on average lower than in the EMAC/SRF simulation, no increase in transpiration is expected. However, as soil moisture is significantly increased in the new coupled model, an increase in evaporation would be plausible and indeed requires a closer analysis. Therefore, evaporation was included in the LST analysis in section 4.1. For further discussion see below (comment 3) .

3. For land surface temperature (LST), it would be interesting to include a discussion about why the latent heat fluxes in EMAC/JSBACH are somewhat overestimated (lines 264-269). For instance, does the overestimated TWS partially contribute to this? Also, latent heat alone may not be sufficient to explain LST. I wonder if other energy fluxes, such as surface shortwave and longwave radiative fluxes, and sensible heat are examined as well.

> Table 3 gives a general overview of the model's performance with respect to other energy fluxes, such as radiative and surface heat fluxes. The comparison of the global mean values of radiative fluxes and heat fluxes shows the largest discrepancy between observation and simulation for the latent heat flux. As a result, the latent heat flux is analyzed in more detail with regard to the lower LST. Since the main driver for the surface latent heat flux is evaporation plus transpiration, indeed the overestimated TWS may have major contribution to the cooler LST.

Evapotranspiration also includes transpiration of vegetation, the latter is not significantly increased compared to the EMAC/SRF simulation and reference data sets. Instead, soil moisture is drastically increased, which most likely affects surface evaporation. Figure 5 was modified, and includes now the evaporation derived from both simulations and reanalysis data of ERA5. The following sentences in line 273 of the revised manuscript:

" Evapotranspiration has a cooling effect on the surface, due to the energy absorbed during the phase change of the water. As a result, cooler LST values are found in regions where evapotranspiration is more intense, such as the tropics and extra-tropics."

 were replaced by:

"Evapotranspiration, the sum of evaporation and transpiration, has in general a cooling effect on the evaporating surface due to energy absorption during the phase change of water. Fig. 5 displays besides LST and latent heat flux, the surface evaporation which is strongest in the tropics and sub-tropics. This is in line with cooler LST values in those regions. The partially overestimated TWS could be the cause of the stronger latent heat flux, as more water is available for evaporation. As the moisture content of the soil in EMAC/JSBACH is in general much larger than in EMAC/SRF, increased evaporation in the coupled simulation is plausible."

4. Section 2.1, consider adding information about soil layers and their depths.

> In Line 78-80  of the revised manuscript the following sentence:  "It provides a complex soil hydrological transport model including percolation and storage of water in several soil depths, which gives a realistic estimate of soil desiccation and corresponding soil temperature and moisture."

  has been replaced by:

  "It provides a complex soil hydrological transport model including percolation and storage of water in several soil depths, reaching down to 9.8m with increasing layer thickness of 0.065m, 0.254m, 0.913m, 2.902m and 5.7m for the first to fifth layers respectively. This gives a realistic estimate of soil desiccation and corresponding soil temperature and moisture."

5. Line 97-108, I wonder if it's possible to include a schematic to demonstrate these processes.

> An schematic is attached below and was added to the Supplement.  The following sentence was added in line 100 of the revised manuscript :

   "An schematic overview of JSBACH as new submodel in EMAC and corresponding process calls is given in the Supplement."

6. Line 161, no values in Table 3 are shown in bold...

> In Table 3 the corresponding values are now displayed in bold.

7. Line 175, are aerosol concentrations prescribed?

> Yes, aerosol concentrations are prescribed. For clarification the following sentence was added in line 180  of the revised manuscript :

" Aerosol concentrations are prescribed for all simulations based on Tanré et al (1997)."
(Tanré, D., Kaufman, Y. J., Herman, M., & Mattoo, S. (1997). Remote sensing of aerosol properties over oceans using the MODIS/EOS spectral radiances. Journal of Geophysical Research: Atmospheres, 102(D14), 16971-16988.)

8. Line 291, the soil depth in the ERA5 is much shallower than the EMAC/JSBACH, 2.89 m vs 9.8 m. How does this affect the comparison of TWS?

➢ It does not affect the TWS comparison, since TWS includes also the runoff. TWS represents all water of the soil per gridbox. This is the main reason why TWS was chosen as evaluation variable.

9. Line 372, "cloud occurrence... remain the same", the differences in LST and latent heat may affect cloud distribution.

➢ Thanks for pointing this out. Since we tune the models top of atmosphere radiation balance by adjusting the cloud characteristics, of course cloud occurrence has changed. Therefore line 387 of the revised manuscript was adjusted to

"Since there are no significant differences between the EMAC/SRF and EMAC/JSBACH surface albedo, no significant differences in Rad_TOA are expected."

10. In terms of TOA fluxes, have you considered using CERES? Or are ERA5 TOA fluxes assimilated with observations?

➢ ERA5 TOA fluxes are assimilated with observations, and to ensure consistency among evaluation variables such as surface albedo, we persisted with the ERA5 dataset.

11. Line 408, are the prescribed SSTs the same in the EMAC/JSBACH and EMAC/SURFACE runs?

➢ Yes, EMAC/JSBACH and EMAC/SRF use the same sea surface temperature and sea ice datasets. In Line 185 of the revised manuscript the following sentence:

" The sea surface temperature and ice concentration is derived from ERA5 six hourly data from 1940 to present (Hersbach et al., 2020) . "

has been replaced by:

" The sea surface temperature and ice concentration is based on ERA5 six hourly data from 1940 to present (Hersbach et al., 2020) and are the same for all performed simulations."

**Technical corrections:**

1. Line 240, 0.1° by 0.1°?

➢ "0.1° x0.1°" has been replaced by "0.1° by 0.1°"

2. Fig. 3 captions, "LST trend" is somewhat misleading as no trends are calculated. Maybe "LST time series"?

➢ In the caption of Figure 3 and Figure 7, the word "trend" has been replaced by "time series".

Schematic (Comment 5) :

[Figure]

**Response to Referee#2**

Thank you very much for the many very useful and constructive comments and suggestions. All suggestions were incorporated into the manuscript and are listed below. The suggestions highlighted in green have been incorporated into the manuscript without further comment, while those requiring extensive discussion are highlighted in black.

**Comments/Remarks/Suggestions:**

Line 2: "the soil water bucket model" ✉ "the soil water bucket model included in the SURFACE submodel"

Line 8: "coupled model" -> "new coupled model (EMAC/JSBACH)"

> ➤ Since the name of the simulation setup is EMAC/JSBACH, labeling the model with the same name would be misleading.

Line 9: "MODIS" ✉ "Moderate Resolution Imaging Spectroradiometer (MODIS)"

Line 15: I would move the sentence "The LAI climatology in EMAC has been substituted with a refined method for directly calculating LAI" to line 6

> ➤ The following sentence was added to line 5 of the revised manuscript:
> " The LAI climatology in EMAC has been substituted with a phenology module calculating the LAI."

Lines 16/17: I would remove here the sentence "FaPAR and GPP exemplify two of the many additional variables made available through JSBACH in EMAC", but insert a sentence in the area of line 4 to 7, describing which variables are now new in EMAC (not included in SURFACE before).

> ➤ Listing all newly available output variables would be too much, since there are more than 300 of them. However a selection of additional output and diagnostic variables was attached in the Supplement in Table TS5 and TS6. The following sentence was added to line 543 of the revised manuscript:
>
> "(a selection of the additional output variables is included in the Supplement)"

Lines 20 to 23: "This improvement can be attributed to a general increase in soil moisture and water storage in deeper soil layers, leading to a reduction in normalised root mean square error (NRMSE) and a closer alignment of simulated TWS with observations, mitigating the previously widespread problem of soil drought." ✉ "This improvement can be attributed to a general increase in soil moisture and water storage in deeper soil layers, and a closer alignment of simulated TWS with observations, mitigating the previously widespread problem of soil drought."

Line 23: I would either remove the sentence "The correlation of TWS and observations is 0.251 and the average global difference is 0.052m water", because that sounds contradictory (due to the very low correlation and the fact that the means are not mentioned) to the previous sentence or alternatively mention also the means 1.13m (model) and 1.078m (ERA5).

Line 24: "We show that the numerous newly added components strongly improve the land hydrology, e.g. soil moisture; while surface parameters, which were mostly prescribed according to climatologies, remain similar." ✉ "We show that the numerous newly added components strongly improve the land

surface, e.g. soil moisture, TWS, and LAI, while surface parameters, as LST, surface albedo or RadTOA, which were mostly prescribed according to climatologies, remain similar."

Line 34: "Gutiérrez et al., 2021". I think this is not the correct citation for the Annex II of IPCC, 2021. The correct citation (also in the literature) is in my opinion an mention in the document itself: IPCC, 2021: Annex II: Models [Gutiérrez, J M., A.-M. Tréguier (eds.)]. In Climate Change 2021: The Physical Science Basis. Contribution of Working Group I to the Sixth Assessment Report of the Intergovernmental Panel on Climate Change [Masson-Delmotte, V., P. Zhai, A. Pirani, S.L. Connors, C. Péan, S. Berger, N. Caud, Y. Chen, L. Goldfarb, M.I. Gomis, M. Huang, K. Leitzell, E. Lonnoy, J.B.R. Matthews, T.K. Maycock, T. Waterfield, O. Yelekçi, R. Yu, and B. Zhou (eds.)]. Cambridge University Press, Cambridge, United Kingdom and New York, NY, USA, pp. 2087–2138, doi:10.1017/9781009157896.016

Line 37: "... (Roeckner et al., 2006)" ✉ "... ,Roeckner et al., 2006)"

Line 37: "However, all physical parameterisations from ECHAM have been replaced ...". I find this sentence misleading, as the parameterizations were not replaced in most cases, but rather outsourced to certain submodels and where appropriate supplemented there with further parameterizations. It is better described in the lines 124/125.

Line 46: "(LPJ-GUESS)" –> "(LPJ-GUESS, Forrest et al., 2020)"

Line 49: "documented" ✉ "described"

Line 51: "(Reick et al., 2021)" ✉ "(Reick et al., 2013, 2021)"

Line 59: "SURFACE" ✉ "the MESSy submodel SURFACE"

Line 67: "document" ✉ "describe"

Line 69: "evaluation" ✉ "corresponding evaluation"

Line 73: "Reick et al. (2013)" ✉ "Reick et al. (2013, 2021)"

Line 75: "Briefly summarized, on the technical side ..." ✉ "On the technical side ..."

Line 86: "The ICON-Land infrastructure allows a clear separation of the physical processes used in JSBACHv4." ✉ "In the case of the ICON-Land infrastructure a clear separation of the physical processes used in JSBACHv4 is allowed."

Line 87: "defined via gross and net primary productivity and photosynthesis" ✉ "defined via gross (GPP) and net primary productivity (NPP) and photosynthesis"

Lines 91/92: Please add a sentence that different land cover types are listed in Table A2.

Line 99: "jsbach" ✉ "JSBACH"

Lines 108/109: I would make a paragraph here (before JSBACH).

Line 129: "In future simulations where JSBACH is used, the SURFACE submodel must be switched off in the namelist setup." ✉ "In the case JSBACH is used, the SURFACE submodel must be switched off."

Line 141: "The simulation based on the default parameters ..." ✉ "The simulation using JSBACH based on the default parameters ..."

Lines 144/145: "... Table S2 of the Supplement. Simulation 2 and 31 were not completed due to server failures and were excluded from the analysis." → "Table S2 of the Supplement (simulations 2 and 31 were not completed due to server failures and were excluded from the analysis)."

Line 148: "ERA5/ERA5-Land monthly averaged data" → "ERA5-Land monthly averaged data"

Lines 160/161: I my opinion you should write two or three sentences more here regarding the results in Table 3. What criteria were applied to decide which parameter sizes were selected?

> In Line 155 of the revised manuscript the following sentence "The optimized parameters that yield the closest fit to the reference data and with the smallest changes were selected based on the lowest normalized root mean square error (NRMSE) sum. "

was replaced by:

"The criteria used for the selection of the optimized tuning parameters were on the one hand the smallest deviation from the reference data paired with the lowest normalized root mean square error (NRMSE) sum and on the other hand the change of as few parameters as possible to stay as close as possible to the tuning of EMAC/SURF."

Line 173: "standard 11 tile setup" → "standard tile setup"

Lines 207/208: I would make a paragraph here (before "LST")

Line 208: "(shown in red in Fig. 1)". In my case this is not red, rather brown. Perhaps don´t mention the colours, in my opinion this is not necessary. I would write "(see Fig.1 and Table 6)".

Line 209: "0.743 K warmer than REF". According to Table 6 it has to be "0.816 K".

Line 211: "0.946" → "0.947" and "0.943" → "0.944" (according to Table 6)

Line 212: "-0.012" –> "-0.013". Probably the reason is the rounding …

Line 213: " Wm-2 " → " Wm-2 "

Lines 239/240: (Muñoz Sabater, 2019, 2021) → "(Muñoz Sabater, 2019)". The both datasets Muñoz Sabater, 2019 and Muñoz Sabater, 2021 were combined into one data set.

Line 246: "is lower everywhere" → "is lower than REF everywhere"

Line 258: "warmer land surface" → "warmer global land surface"

Line 261: "tropics, subtropics" → "tropics/subtropics"

Line 262: "in both simulations" → "in the EMAC/JSBACH and the EMAC/SRF simulation"

Line 283: Also evaporation?

> In Line 295 of the revised manuscript the following sentence was added:
> "TWS does not include evaporation."

Line 293: "In Fig. 7 the difference of TWS between EMAC/JSBACH and ERA5 is shown." ✉ "In Fig. 7 the difference of TWS between EMAC/JSBACH and EMAC/SRF to ERA5 is shown."

Line 293ff, Fig.6/Fig.7: I would change the numbers of the two figures, Figure 7 to Figure 6 and Figure 6 to Figure 7, so that they appear in the text in the correct order.

Line 295: "Russia" ✉ "Western Russia"

Line 295: "EMAC/JSBACH overestimates TWS almost everywhere, except for high elevated regions" ✉ "EMAC/JSBACH overestimates TWS almost everywhere, independent of the season, except for high elevated regions"

Line 298: ✉ Fig.7 (right panels)

Line 299: "... than the EMAC/SRF results, which is also visible in Fig. 6." ✉ "... than the EMAC/SRF results. This is also visible in Fig. 6. where the globally averaged TWS trend is illustrated."

Line 300: "The TWS of the EMAC/SRF simulation is lower everywhere ..." ✉ "The TWS of the EMAC/SRF simulation (Fig. 7) is lower everywhere ..."

Line 302: "−0.68 m lower than the one derived from reanalysis data." ✉ "−0.684 m lower than the one derived from reanalysis data (see Table 6). In EMAC/JSBACH the global average of TWS is 1.13 ± 0.706 m which is, with a difference of 0.052m, significantly closer to ERA5 (1.078±0.56m)."

Line 304: "computed" ✉ "simulated"

Line 310: "0 ∘C" ✉ "0 °C"

Line 325: I would make a paragraph here (after "... (2021).")

Line 327: "?Schaaf and Wang," ✉ "Schaaf and Wang,"

Lines 330-332: "During summer, EMAC/JSBACH shows a slight overestimation in the Northern Hemisphere and an underestimation in the Southern Hemisphere. In the winter months the opposite applies, underestimation in the Northern Hemisphere and overestimation in the Southern Hemisphere." This is only true for different parts of the Northern Hemisphere and Southern Hemisphere. Please be more specific.

> The following sentence was added in line 345 of the revised manuscript :
> "During summer, EMAC/JSBACH shows a slight overestimation over Europe and Asia between 45 and 75°N, and parts of Canada. Between 25°S and 15°N an underestimation is visible. During the northern winter months North America and Canada show underestimated surface albedo as well as Scandinavia, Eastern Europe, Northern Russia and Elevated Regions in Asia.

Line 336: "Fig.8" ✉ "Fig.8 (right panels)"

Line 355: "0.907" ✉ "0.907 (Table 6)"

Line 355: "ERA5 and EMAC/JSBACH" ✉ "EMAC/JSBACH and ERA5

Line 356: "... of EMAC/JSBACH RadTOA during ... " ✉ ""... of EMAC/JSBACH RadTOA in comparison to ERA5 during ..."

Line 370: "correlation" ✉ "anti-correlation"

Line 387ff/Fig.10/Fig.11: I would also here change the numbers of the two figures, Fig. 10 to Fig. 11 and Fig. 11 to Fig.10, so that they appear in the text in the correct order.

Lines 388: "… in the simulation results in comparison …" ✉ " … in the EMAC/JSBACH simulation in comparison …"

Line 390: "… is 0.042 mm day−1. " ✉ "… is 0.042 mm day-1 (2.738 to 2.696 mm day-1 , see Table 6)." This makes it easier to classify the size of the difference.

Lines 427/428: : I would make a paragraph here (after "… grid.")

Line 427: "… is −0.212 m2 m−2 lower …" ✉ "is −0.212 m2 m−2 (1.187 to 1.399, see Table 6) lower …"

Line 431: "The zonal average shows" –> "The zonal averages (Fig. 12 right panels) shows"

Line 516: "0.056% ± .4e − 05%" ✉ "0.056% ± 4e−04%

Line 518: "Results indicate that the LST derived from the newly coupled model is on average 1.546∘K colder compared to the LST derived from ERA5." ✉ "Results indicate that the LST derived from the newly coupled EMAC/JSBACH model is on global average 1.546 K colder compared to the LST derived from ERA5 (using the old SURFACE submodel, the globally averaged LST was 0.816 K warmer)."

Line 520: "the reanalysis" ✉ "the ERA5 reanalysis"

Line 525: "are among many other newly introduced variables". I would list them all here.

> See above (comment on Lines 16/17)

Table 1: "NPP" ✉ "Net primary productivity (NPP)"

Table 3: In the case that the abbreviation of a physical quantity (e.g. HFLXsensible) are not explained (long form) in the main text, explain them here at least in the caption

> The following was added to the caption of Table 3:
> "TOAnet refers to the sum of shortwave (TOAsw) and long-wave (TOAlw) top of atmosphere radiation flux, while SRF* refers to the same at surface level. HFLXnet refers to the sum of the sensible (HFLXsensible) and latent (HFLXlatent) heat flux. Clouds are assessed based on the accumulated cloud cover (ACLC), the liquid water content (LWC) and the ice water content in clouds (IWC)."

Table 6:
• For the precipitation there is the value 2.738 ± 3.382 (EMAC/JSBACH) and 3.025 ± 3.279 (EMAC/SRF). That cannot be, or? The standard deviation is larger than the mean. This would lead to negative precipitation values …

> There was a mistake in the calculation of the standard deviation, which has been corrected for all variables in Table 6.

Table A1: Why do you have two times the number 1 and two times the number 11 here?

> Land cover type (lct) 01 and 02 share a tile, this is possible since lct01 (glaciers) never appears on gridboxes of lct02 (tropical evergreen forest), the same is true for lct15 and lct16. This approach saves computational resources.

Table S2 (Supplement):

- It is not totally clear that the numbers in the EMAC/SRF and in the CTRL rows are the default values. Maybe you can write this in the caption. Furthermore, if the value appears in the table, it would also be good to write "default" there. For example in run 4 instead of "0.85" for zasic write "default". Perhaps the latter is also sufficient to make it clear.
- Please also write in the caption that the simulations 2 to 35 were performed with EMAC/JSBACH.

    ➢ Table S3 and S4 were adjusted accordingly (caption and content).

- What is the difference between CTRL and the runs 4, 8, 20, and 22? Maybe I don´t see it, but are these not all the simulation setups?  In Table S3 and S4 are the same results in the case of the simulations 4, 8, 20 and 22. But the results differ to the CTRL simulation. Why is this so?

    ➢ There is no difference between simulations 4, 8, 20 and 22. However, this was only discovered when the results were analyzed, nevertheless the simulations are listed to provide a complete record.  The values of the CTRL simulation in Table S3 S4 have been corrected.

- Why do you write "EMAC/JSBACH" instead of "16" in the case of run 16. That is confusing. If you want to express that this is the best choice of parameters, please also write this in the caption.

    ➢ Table S2, S3 and S4 have been adjusted accordingly. Each simulation now has a number and their specific names are added in brackets.

Fig. 5:

- "low cloud cover (lcc), medium cloud clover (mcc) and high cloud cover (hcc)" are not shown.
- "The blue background color indicates values averaged over the polar climate zone (latitudes > 66.5 ∘), the green background color indicates values averaged over the temperate climate zone (latitudes between 40 ∘ and 66.5 ∘), and the red background color indicates values averaged over the tropical and subtropical climate zone (latitudes < 40 ∘)" "The upper panels indicates values averaged over the polar climate zone (latitudes > 66.5 ∘), the mid panels values averaged over the temperate climate zone (latitudes between 40 ∘ and 66.5 ∘), and the bottom panels values averaged over the tropical and subtropical climate zone (latitudes < 40 ∘)."

    ➢ The figure description was adjusted accordingly.

References:

1) "Gutiérrez, J M., A.-M. T. e., Masson-Delmotte, V., P. Z. A. P. S. C. C. P. S. B. N. C. Y. C. L. G. M. G. M. H. K. L. E. L. J. M. T. M.-T. W.O. Y. R. Y., and (eds.), B. Z.: Annex II: Models, p. 2087–2138, Cambridge University Press, Cambridge, United Kingdom and New York, NY, USA„ https://doi.org/10.1017/9781009157896.016, 2021"

[Figure]

„IPCC, 2021: Annex II: Models [Gutiérrez, J M., A.-M. Tréguier (eds.)]. In Climate Change 2021: The Physical Science Basis. Contribution of Working Group I to the Sixth Assessment Report of the Intergovernmental Panel on Climate Change [Masson-Delmotte, V., P. Zhai, A. Pirani, S.L. Connors, C. Péan, S. Berger, N. Caud, Y. Chen, L. Goldfarb, M.I. Gomis, M. Huang, K. Leitzell, E. Lonnoy, J.B.R. Matthews, T.K. Maycock, T. Waterfield, O. Yelekçi, R. Yu, and B. Zhou (eds.)]. Cambridge University Press, Cambridge, United Kingdom and New York, NY, USA, pp. 2087–2138, doi:10.1017/9781009157896.016"

2) "Hersbach, H., B. B. B. P. B. G. H. A. M. S. J. N. J. P. C. R. R. R. I. S. D. S. A. S. C. D. D. T. J.-N.: ERA5 monthly averaged data on singlelevels from 1940 to present, Copernicus Climate Change Service (C3S) Climate Data Store (CDS), https://doi.org/10.24381/cds.f17050d7,615, 2023."

[Figure]

"Hersbach, H., Bell, B., Berrisford, P., Biavati, G., Horányi, A., Muñoz Sabater, J., Nicolas, J., Peubey, C., Radu, R., Rozum, I., Schepers, D., Simmons, A., Soci, C., Dee, D., Thépaut, J-N. (2023): ERA5 monthly averaged data on single levels from 1940 to present. Copernicus Climate Change Service (C3S) Climate Data Store (CDS), DOI: 10.24381/cds.f17050d7 (Accessed on DD-MMM-YYYY)"

3) "Muñoz Sabater, J.: ERA5-Land monthly averaged data from 1981 to present, https://doi.org/10.24381/cds.68d2bb3, 2019.

Muñoz Sabater, J.: ERA5-Land monthly averaged data from 1950 to 1980, https://doi.org/10.24381/cds.68d2bb3, 2021"

[Figure]

"Muñoz Sabater, J. (2019): ERA5-Land monthly averaged data from 1950 to present. Copernicus Climate Change Service (C3S) Climate Data Store (CDS). DOI: 10.24381/cds.68d2bb30 (Accessed on DD-MMM-YYYY)"